# Comprehensive organic emission profiles for gasoline, diesel, and gas-turbine engines including intermediate and semi-volatile organic compound emissions

Quanyang Lu [1, 2], Yunliang Zhao [1, 2, 3], Allen L. Robinson [1,2]

[1]Department of Mechanical Engineering, Carnegie Mellon University, Pittsburgh, Pennsylvania 15213, United States
[2]Center for Atmospheric Particle Studies, Carnegie Mellon University, Pittsburgh, Pennsylvania 15213, United States
[3]Now at: California Air Resources Board, Sacramento, California 95814, United States

*Correspondence to*: Allen L. Robinson (alr@andrew.cmu.edu)

**Abstract.** Emissions from mobile sources are important contributors to both primary and secondary organic aerosols (POA and SOA) in urban environments. We compiled recently published data to create comprehensive model-ready organic emission profiles for on- and off-road gasoline, gas-turbine, and diesel engines. The profiles span the entire volatility range, including volatile organic compounds (VOCs, effective saturation concentration $C^*=10^7$-$10^{11}$ µg/m$^3$), intermediate-volatile organic compounds (IVOCs, $C^*=10^3$-$10^6$ µg/m$^3$), semi-volatile organic compounds (SVOCs, $C^*=1$-$10^2$ µg/m$^3$), low-volatile organic compounds (LVOCs, $C^*\leq0.1$ µg/m$^3$) and non-volatile organic compounds (NVOCs). Although our profiles are comprehensive, this paper focuses on the IVOC and SVOC fractions to improve predictions of SOA formation. Organic emissions from all three source categories feature tri-modal volatility distributions ('by-product' mode, 'fuel' mode, and 'lubricant oil' mode). Despite wide variations in emission factors for total organics, the mass fractions of IVOCs and SVOCs are relatively consistent across sources using the same fuel type; for example, contributing 4.5% (2.4-9.6% as 10$^{th}$ to 90$^{th}$ percentile) and 1.1% (0.4-3.6%) for a diverse fleet of light duty gasoline vehicles tested over the cold-start unified cycle, respectively. This consistency indicates that limited number of profiles are needed to construct emissions inventories. We define five distinct profiles: (i) cold-start and off-road gasoline, (ii) hot-operation gasoline, (iii) gas turbine, (iv) traditional diesel and (v) diesel-particulate-filter equipped diesel. These profiles are designed to be directly implemented into chemical transport models and inventories. We compare emissions to unburned fuel; gasoline and gas-turbine emissions are enriched in IVOCs relative to unburned fuel. The new profiles predict that IVOCs and SVOC vapor contribute significantly to SOA production. We compare our new profiles to traditional source profiles and various scaling approach used previously to estimate IVOC emissions. These comparisons reveal large errors in these different approaches ranging from failure to account for IVOC emissions (traditional source profiles) to assuming source-invariant scaling ratios (most IVOC scaling approaches).

# 1 Introduction

Atmospheric particulate matter imposes health risks (Di et al., 2017) and influences climate (Kanakidou et al., 2005). Organic aerosol (OA) contributes 20-90% of submicron atmospheric fine particulate matter mass (Jimenez et al., 2009). OA is commonly classified as primary OA (POA), which is directly emitted by sources, or secondary OA (SOA), which is formed
in the atmosphere through photo-oxidation gas-phase organics. Both POA and SOA concentrations depend on the gas-particle partitioning of a complex mixture of organics that span a broad range of volatility (Hallquist et al., 2009; Kroll and Seinfeld, 2008). Mobile sources contribute about one-third of the anthropogenic organic emissions in the 2014 EPA National Emission Inventory (NEI); they are an important source of POA and SOA precursor gases especially in urban environments (Gentner et al., 2017; USEPA-OAQPS, 2015).

Traditional emissions inventories such as the NEI account for emissions of gas-phase volatile organic compounds (VOCs, typically smaller than $C_{12}$) and non-volatile particulate matter (PM). These emissions are speciated for use in chemical transport models using source-specific emission profiles. Robinson et al. (2007) and Shrivastava et al. (2008) argued that this is an overly simplistic representation of organic emissions.

First, multiple studies have demonstrated that a large fraction of POA is semi-volatile with dynamic gas-particle partitioning
while traditional inventories and models treat it as non-volatile (Fujitani et al., 2012; Kuwayama et al., 2015; Li et al., 2016; May et al., 2013b, 2013a, 2013c; Robinson et al., 2007). Semi-volatile POA concentrations depend on the gas-particle partitioning of the emissions, which is determined by their volatility distribution and atmospheric conditions. In addition, source tests are often conducted at unrealistically high OA loading, which biases POA emission factor compared to more dilute, atmospheric conditions (Fujitani et al., 2012; Lipsky and Robinson, 2006). Second, most traditional inventories do not
account for emissions of lower volatility organic gases, including intermediate-volatile organic compounds (IVOCs, effective saturation concentration $C^*=10^3-10^6$ µg/m$^3$) and semi-volatile organic compounds (SVOCs, $C^*=1-10^2$ µg/m$^3$). Laboratory experiments indicate that IVOCs and SVOCs form SOA efficiently (Chan et al., 2009; Presto et al., 2010), but quantifying their emissions requires sorbents which are not routinely used for source testing (Kishan et al., 2008). Neglecting SOA production from IVOCs and SVOCs can lead to substantial underprediction of atmospheric SOA production (Hodzic et al.,
2010; Woody et al., 2016). The net effect of these two issues is to cause chemical transport models to overestimate POA emissions and underestimate SOA production, leading to errors in the predicted OA composition and concentrations (Baker et al., 2015; Ensberg et al., 2014; Woody et al., 2016). Accounting for these two issues improves model-measurement agreement (Jathar et al., 2017; Murphy et al., 2017; Woody et al., 2016).

IVOC and SVOC emissions have not been routinely implemented in models because of lack of the mass and chemical
composition of total IVOCs and SVOCs (Shrivastava et al., 2008). Although many studies report emissions of individual IVOC and SVOC species (typically polycyclic aromatic hydrocarbon or *n*-alkanes) (Schauer et al., 1999a, 1999b, 2002; Siegl et al., 1999; Zielinska et al., 1996), the vast majority of the IVOC/SVOC mass cannot be resolved at the molecular level using traditional gas chromatography based techniques (Goldstein and Galbally, 2007; Zhao et al., 2014).

Recent studies have reported comprehensive IVOC, SVOC and/or low-volatile organic compound (LVOC, $C^* \leq 0.1$ µg/m$^3$) emissions and gas-particle partitioning on POA emissions from mobile sources (May et al., 2014; Presto et al., 2011; Zhao et al., 2015, 2016). Zhao et al. (2015, 2016) characterized the total emissions and chemical composition of IVOCs and SVOCs from a fleet of on- and off-road gasoline and diesel sources. Cross et al. (2013, 2015) reported total IVOC and/or SVOC emission from an aircraft and diesel engine. Presto et al. (2011) and Drozd et al. (2012) reported IVOC and SVOC emissions for two gas-turbine engines. Gentner et al. (2012) and Isaacman et al. (2012a) report molecular and mass spectrum information for IVOC and SVOC in liquid fuel and quartz filter samples. May et al. (2013a, 2013b), Kuwayama et al. (2015), and Li et al. (2016) also investigated the gas-particle partitioning of on-road vehicle POA in dynamometer and tunnel studies. However, only limited comparisons have been made between source categories and the data have not been compiled into model ready profiles.

In this paper, we report comprehensive organic emission profiles for mobile sources by integrating recently published data of organic emissions based on their volatility, including IVOCs and SVOCs, to improve model predictions of SOA formation. We compare our new profiles to traditional source profiles and unburned fuel, focusing on the volatility distribution and SOA precursors. We then use the new profiles to evaluate different scaling approaches previously used to incorporate IVOC emissions into inventories and models. Finally, we present box model calculations of SOA formation to demonstrate the importance to implement the new profiles in SOA modelling.

## 2 Methods

### 2.1 Datasets

This paper combines previously published measurement data of organic emissions (Gordon et al., 2013; May et al., 2014; Presto et al., 2011; Zhao et al., 2015, 2016) from gasoline, gas-turbine and diesel engines to create comprehensive model-ready source profiles. All tests used the same procedures to characterize IVOC and SVOC emissions to create a self-consistent dataset for low-volatile organics, but slightly different sampling media (Tedlar bags and/or canisters) and level of speciation were used to characterize VOC emissions. In the results and discussion sections, we compare these data to other recently published measurements made using different techniques.

We present two types of data: (i) emission factors of total organics and (ii) speciation profiles. We present total organic emissions factors for all tested engines: 64 gasoline vehicles, 5 diesel trucks, 6 off-road gasoline engines, 1 off-road diesel engine and 1 gas-turbine engine. We define total organic emissions as the sum of non-methane organic gases (NMOG) measured by flame ionization detection plus 1.2 times organic carbon (OC) measured using thermal optical analysis of quartz filter sample (the factor of 1.2 is the organic-mass-to-organic-carbon ratio, which accounts for the contribution of non-carbonaceous species in the organic (Turpin and Lim, 2001)). We define the NMOG as THC (measured with FID) minus CH$_4$ plus carbonyls. We define POA as organics collected by a bare quartz filter analyzed by thermal-optical analysis. We converted measured pollutant concentrations to fuel-based emission factors (EF, mg/kg-fuel) using the carbon-mass-balance approach

and the measured mass fraction of carbon in fuel (0.82 for gasoline, 0.86 for jet fuel and 0.85 for diesel) (May et al., 2014; Presto et al., 2011).

We derive speciation profiles from gas-chromatography-based analyses of filter, adsorbent tubes and Tedlar bag/canister samples. Details on the analytical procedures are described by Zhao et al. (2015, 2016). The speciation profiles are based the

subset of tests with complete data (all three media): VOCs, IVOCs, SVOCs, and LVOCs. This included 29 gasoline vehicles, 4 diesel trucks, 3 off-road gasoline engines, 1 off-road diesel engine and 1 gas-turbine engine (Table S1). A detailed description of experimental set-up, sampling and chemical analysis is provided in the original articles (Gordon et al., 2013; May et al., 2014; Presto et al., 2011; Zhao et al., 2015, 2016). Only a brief description is provided here.

Emissions samples were collected from diluted exhaust. Gasoline and diesel source emissions were collected from a constant

volume sampler (CVS) that diluted the exhaust with ambient air treated by high-efficient particulate air (HEPA) filters (Gordon et al., 2013; May et al., 2014). Gas-turbine engine exhaust was sampled from a rake inlet installed 1-m downstream of the engine exit plane (Presto et al., 2011). Sources were tested using standard test cycles (Gordon et al., 2013; May et al., 2014; Presto et al., 2011). On-road gasoline vehicles were tested on both cold-start and hot-start unified cycles. On-road diesel vehicles were tested in both lower-speed (creep and idle) and high-speed operation modes. Gas-turbine engine was operated

on 4% and 85% engine thrust. Off-road engines were operated on certification cycles.

A suite of complementary sampling media was employed to characterize emissions across the entire volatility range. Tedlar bags (for gasoline and diesel sources) or canisters (for gas-turbine source) were collected and analyzed by GC-FID and GC-MS to determine $CH_4$ and VOC hydrocarbon emissions up to $C_{12}$ compounds (May et al., 2014; Presto et al., 2011). Carbonyls (up to $C_6$) were sampled using 2,4-dinitrophenylhydrazine (DNPH) impregnated cartridges and analyzed by high-performance

liquid chromatography (HPLC) (May et al., 2014). Quartz filters followed by two Tenax TA adsorbent tubes collected low-volatility organics that were analyzed by GC/MS equipped with a thermal desorption and injection system (Gerstel) (Zhao et al., 2015, 2016). The filter samples were also analyzed using a Thermal/Optical Carbon Analyzer for total organic carbon (OC) (May et al., 2014). The adsorbent tubes collect IVOCs and some SVOCs; SVOCs and even lower volatility organics were collected on quartz filters (Zhao et al., 2015, 2016). Except for the gas-turbine engine tests, total hydrocarbon (THC) emissions

were determined by FID analysis of Tedlar bag samples (Gordon et al., 2013; May et al., 2014).

All adsorbent tubes and quartz filters were analyzed following the same procedure. Total (speciated and unspeciated) mass of IVOCs, SVOCs and LVOCs was determined by Zhao et al. (2015, 2016). The analysis quantified 57 individual IVOCs, which together contributed less than 10% of the total IVOC mass. The residual IVOCs, SVOCs and LVOCs commonly appear as an unresolved complex mixture (UCM); they were quantified into 29 lumped group ($C_{12} - C_{38}$) based on the retention time of *n*-

alkanes (each group corresponds to the mass that elutes between two sequential *n*-alkanes). Each IVOC lumped group ($C_{12} - C_{22}$) was further subdivided into two chemical classes (unspeciated branched and cyclic compounds) based on their mass spectra. NVOCs are determined as the difference between the thermal optical analysis (1.2*OC) and the GC/MS analysis (IVOC+SVOC+LVOC) of the quartz filter samples.

Different levels of speciation were performed on the canister or Tedlar bag samples, depending on source category. The Tedlar bag samples of gasoline exhaust were analyzed for 192 individual VOCs and 10 IVOCs; gas turbine exhaust was analyzed for 81 individual VOCs and 5 IVOCs; diesel exhaust was analyzed for 47 individual VOCs, 2 IVOCs and 11 Kovats lumped groups in the VOC range (organics with a GC retention time between $n^{th}$ and $n+1^{th}$ $n$-alkanes). Given the different levels of VOC characterization, we supplemented our gas-turbine and diesel VOC data with existing speciation profiles (SPECIATE profiles #4674 and #5565). The method for combining the VOC data is described in Supporting Information.

## 2.2 Mapping organics into volatility basis sets

Gas-phase organic emissions must be speciated for use in chemical mechanisms such as SAPRC (Carter, 2010) or Carbon Bond (CB). These mechanisms typically group individual VOCs into a set of lumped compounds based on reactivity or other chemical properties. We compared gas-phase organic emissions using the lumping specified by the SAPRC mechanism; we also compare gas- and particle-phase emissions using the volatility basis set (VBS). The VBS framework lumps organics into logarithmically spaced bins of saturation concentrations (C*) at 298K. It is designed for representing the emissions and atmospheric evolution of lower volatility organics ($C_{12}$ and larger) in chemical transport models (Donahue et al., 2006). It is also useful visualizing and comparing emissions data across the entire volatility space; the VBS is not intended to replace chemical mechanisms used to represent VOCs in models. Figure S1 shows the overall processes of mapping speciated and unspeciated compounds data collected on sampling medias to volatility basis set (VBS).

To map emissions into the VBS, we assigned C* values to individual compounds and lumped groups of unspeciated organics. For each speciated compound (i.e. individual VOCs and IVOCs), C* values are calculated as,

$$C_i^* = \frac{M_i 10^6 \zeta_i p_{L,i}^0}{760RT} \quad (1)$$

where $M_i$ is the molecular weight (g/mol), $\zeta_i$ is the activity coefficient of compound $i$ in the condensed phase (assumed to be 1), and $p_{L,i}^0$ is the liquid vapor pressure (Torr) of compound $i$, R is the ideal gas constant ($8.206 \times 10^{-5}$ m$^3$ atm mol$^{-1}$ K$^{-1}$), T is temperature (K). $p_{L,i}^0$ values are from EPA Suite data at 298K (USEPA, 2012). Although experimental and/or predicted vapor pressure values are uncertain (Komkoua Mbienda et al., 2013), the factor of 10 spacing of the volatility bins in the VBS reduces the chance of misclassification errors.

For unspeciated organics, C* values were assigned to lumped groups using the retention time of $n$-alkanes as reference species. In the VOC range, Kovats groups are assigned the mean of log C* value of the two $n$-alkanes in each group (Presto et al., 2012). For IVOCs, SVOCs and LVOCs, the C* value of the $n$-alkane in each bin is used to represent the UCM that elutes around that $n$-alkane. IVOCs, SVOCs and LVOCs correspond to the retention time range of $C_{12}$ to $C_{22}$, $C_{23}$ to $C_{32}$, and $C_{33}$ to $C_{36}$ $n$-alkanes, respectively. Although calibrating C* using $n$-alkanes can overestimate the volatility of PAHs and aromatic oxygenates (Presto et al., 2012), these compounds are expected to contribute only a small fraction of the total low-volatile organics. In addition, the VBS volatility bins are a factor of 10 apart, which reduces the chance of misclassification errors.

After assigning C* values, we compile all species into the VBS volatility distribution. Each volatility bin of $C^* = 10^n \, \mu g/m^3$ cover the volatility range from $C^* = 0.3 \times 10^n \, \mu g/m^3$ to $C^* = 3 \times 10^n \, \mu g/m^3$ in a logarithmic space with n varying from -2 to 11. One challenge is that the Tedlar bags/canister samples were collected in parallel to the filter/adsorbent tubes, which creates concerns about double counting. We assessed this issue by comparing volatility of organics measured by both approaches.

Three IVOC species were measured in both the Tedlar bags and adsorbent samples: *n*-pentyl-benzene ($C^*=2.8 \times 10^6 \, \mu g/m^3$), *n*-dodecane ($C^*=1.9 \times 10^6 \, \mu g/m^3$) and naphthalene ($C^*=1.1 \times 10^6 \, \mu g/m^3$). Figure S2 (a-c) compares the emissions of these species measured using the two approaches (Supporting Information). For the most volatile of these species, n-pentyl-benzene, the Tedlar bag measured, on average, 5.2 times more than the adsorbent tubes. Both approaches measured essentially the same amount of n-dodecane (ratio of 0.85 and $R^2$ of 0.9. For naphthalene (the least volatile of these species), the adsorbent tubes

measured about 5 times more than the Tedlar bag, which we attribute to wall losses in the bag (Wang et al., 1996).

The comparisons indicate that the filter/adsorbent tube sampling train quantitatively collects all organics less volatile than *n*-dodecane ($C^* = 1.9 \times 10^6 \, \mu g/m^3$) while the bag/canister quantitatively collects all more volatile organics. *N*-dodecane falls within the $10^6 \, \mu g/m^3$ volatility bin. The upper bound of this bin is $3 \times 10^6 \, \mu g/m^3$, which is close to the C* of *n*-dodecane. We therefore use $3 \times 10^6 \, \mu g/m^3$ as the boundary between the adsorbent tube and Tedlar bag samples. To avoid double counting, we discarded

all organics measured using the bag/canister/cartridge that are less volatile than $3 \times 10^6 \, \mu g/m^3$ and discarded all species measured in the adsorbent tube more volatile than $3 \times 10^6 \, \mu g/m^3$. Therefore, emissions in the $C^* = 10^7$ to $10^{11} \, \mu g/m^3$ bins are based on the bag/canister/cartridge data and that the emissions in the $C^* = 10^{-1}$ to $10^6 \, \mu g/m^3$ bins are based on the filter and adsorbent tube data. NVOCs are assigned to a non-volatile bin. The adsorbent tubes may underestimate the speciated emissions in C* between $1.9 \times 10^6$ (*n*-dodecane) and $3 \times 10^6 \, \mu g/m^3$; however, they still measured, on average, 3.3 times organics in this

range to the Tedlar bags (Figure S2d).

A final issue is whether our sampling and analytical methods capture and recover all emitted organics. We evaluated this by comparing the sum of total characterized organics (integrated organics from bag, adsorbent tube and filter measurements) to our estimate of total organics by bulk measurements (NMOG+1.2*OC). The sum of the characterized organics includes the VOCs, IVOCs, SVOCs, LVOCs determined from the GC-based analysis of the bags/canister, cartridges, adsorbent tubes and

filters. This includes both individual species and lumped groups of unspeciated material.

Figure S3 indicates good mass closure for the on-road gasoline and diesel vehicle tests. The two estimated results agree within ±10% for more than 90% of non-DPF-equipped diesel engine tests (DPF = diesel particulate filter). For all LDGV (light-duty gasoline vehicle) tests, total characterized organics are 82 ± 21% of the total organics by bulk measurements. We suspect that most of the missing organics from the LDGV tests could be VOCs since the VOC analysis only quantified a list of targeted

compounds (Zhao et al., 2017). There was relatively poor mass closure for the off-road engine and DPF-equipped diesel tests. For the off-road engine emissions, the sum of total characterized organics was less than 50% of the bulk measurement. Comparisons with literature data (Gabele, 1997; Volckens et al., 2008) suggests that our speciated VOC groups to NMOG ratios are low (Figure S4). The cause of this bias is not known, but we attribute it to measurement error. We used a linear regression to the literature results to rescale our VOC data for off-road engines (see SI). For DPF-equipped diesel vehicles, the

sum of speciated organics is up to 7 times the bulk measurement of total organics. The DPF-equipped diesel emission are quite low and this discrepancy is likely due to uncertainty in background corrections (Zhao et al., 2015).

Traditionally, there are three standard ways to treat these residual emissions -- the difference between sum of characterized emissions and the total/bulk emissions (frequently called unknown or UNK): (1) assume it is inert and therefore ignored in
models, (2) renormalizing the residual emissions to the known composition which assumes that the composition of the unspeciated material is the same as the speciated mass, or (3) by assigning a custom profile to the residual mass based on a representative list of compounds (Carter, 2015). The standard default profile for (3) was derived from the all-profile-average carbon number > 6, molecular weight > 120 compounds in SPECIATE database (Adelman et al. 2005). Therefore, it lacks comprehensive IVOCs and SVOCs data.

In the following discussion, we normalize the residual/uncharacterized organics to the known composition, assuming that the residual unknown organics have the same volatility and chemical characteristics as the total characterized organics. Since there was not an independent measurement of NMOG during the gas-turbine engine tests (Presto et al., 2011), we assume the supplemented speciated VOCs plus the sorbent and filter data is the total emitted organics.

## 2.3 Box model for SOA yield calculation

The overall SOA yield of gas-phase emissions (mass of SOA produced/mass of NMOG emissions) can be calculated as

$$y_{SOA} = \sum_i f_{gas,i} \times Y_i \qquad (2)$$

where $f_{gas, i}$ is the mass fraction of SOA precursor $i$ in NMOG; and $Y_i$ is the SOA mass yield of compound $i$ at OA= 10 µg/m$^3$ (a typical urban OA level).

SOA mass yields for each VOCs are based on SAPRC groups and are taken from CMAQ 5.1 (USEPA, 2016a). The complete
VOC composition for the new source profiles are listed in the Table S3. SOA mass yields for IVOCs are calculated using the mechanism of Zhao et. al (2015). The gas-phase SVOCs are assumed to have a SOA mass yield of 1 (Presto et al., 2010). Equation (2) omits the OH reaction rates and therefore represents the ultimate SOA yield from NMOG emissions. The relative contribution of IVOCs and VOCs to SOA varies with time because IVOCs generally react faster with OH than VOCs (Zhao et al., 2016). Therefore, the ultimate yield approach (equation 2) provides a lower bound estimate of the contribution of IVOCs
to SOA.

## 3. Results and discussion

Figure 1 shows the volatility distribution of the total characterized organic emissions for a typical gasoline (Figure 1a) and diesel (Figure 1b) test classified by collection media. It underscores the importance of using adsorbents (in addition to filters and Tedlar bags) to comprehensively characterize all of the organic emissions. The Tenax adsorbent tubes collect almost all
of the IVOCs (> 90% for gasoline and > 97% for diesel), with the balance being collected by the quartz filter, presumably as adsorption artifact (Zhao et al., 2015, 2016). The Tenax adsorbent collects 5.2% and 54.8% of the total organic emissions from the gasoline and diesel engines, respectively. Since the vast majority of source testing does not employ adsorbents, IVOCs are

not quantitatively accounted for in most emission profiles (Pye and Pouliot, 2012). In comparison to the adsorbent samples, the bag/canister collected only 12.9% and 4.0% of IVOCs for gasoline and diesel sources, respectively. We have discarded this component to avoid double counting, as discussed in the methods section.

Figure 1 also shows the particle fraction ($X_p$) calculated assuming all organics form a quasi-ideal solution to illustrate gas-particle partitioning at typical atmospheric conditions (T=298K, OA=10 µg m$^{-3}$). At these conditions, IVOCs exist essentially exclusively in the gas-phase, while SVOCs exist in both phases. To illustrate the changes in gas-particle partitioning of IVOCs and SVOCs across a wide range of atmospheric conditions, Figure S5 shows equilibrium particle fraction ($X_p$) for T between 273K – 320K and OA concentration from 1 – 10 µg m$^{-3}$. IVOCs are essentially exclusively in the gas-phase (>99%) except at very low temperature (T = 273K) and high OA loading (OA = 10 µg m$^{-3}$) conditions when about 6% of the lowest bin (C* = 10$^3$ µg m$^{-3}$) partitions to particle-phase. In contrast, SVOCs are always present in both gas- and particle- phases, in both hot and dilute (T = 320K and OA = 1 µg/m$^3$) or cold and high OA loading (T = 273K and OA = 10 µg m$^{-3}$) conditions.

Figure 1 indicates there is also substantial breakthrough of SVOCs from the quartz filter during mobile source certification testing (e.g. 2007 CFR 86), which requires maintaining a filter temperature of 47°C. In our experiments, these SVOCs are collected by the downstream Tenax tubes. This breakthrough is denoted by the white bars in the SVOC range in Figure 1; the SVOC breakthrough corresponds to, on average, 37% of the total SVOC emissions from gasoline vehicles, 52% for non-DPF diesel and 89% for DPF-diesel (Zhao et al., 2015, 2016). Therefore, quantitatively accounting for all gas-phase SVOCs requires using adsorbents. This is needed to improve predictions of POA concentrations and SOA production.

We compared the sum of NVOCs, LVOCs and SVOCs to the quartz filter POA measurements. A linear regression of the on-road gasoline vehicle data yields a slope of 1.4 (Figure S6a), which indicates that the quartz-filter-based POA emission factors should be multiplied by 1.4 to account for missing gas-phase SVOC emissions. This factor would be larger if the quartz filter did not collect some IVOC vapors as adsorption artifact (Figure 1). For off-road gasoline sources, a linear regression yields a slope of 1.1 (Figure S6b), indicating a larger fraction of the SVOC are collected on quartz filters compared to on-road gasoline vehicles. This is presumably due to shifts in gas-particle partitioning towards the particle-phase at the high OA concentrations in the off-road source tests. For diesel sources, a linear regression yields a slope of 0.9 (Figure S6c). This lower ratio is due to that filter measured POA also includes positive adsorption artifact from IVOCs, which more than offsets the gas-phase SVOC breakthrough (Figure 1b).

### 3.1 Emission factors

Figure 2(a) compares the total organics emission factors (NMOG+1.2*OC) for on- and off-road gasoline vehicles, including LDGV, two-stroke small off-road engines (SORE-2S), and four-stroke small off-road engines (SORE-4S); gas-turbine engines; and on- and off-road diesel sources, including DPF-equipped engines. We subdivided the LDGV data based on emissions certification standard: pre-LEV (U.S. Tier0), LEV (California Low Emission Vehicle), and ULEV (California Ultra-Low Emission Vehicle).

Although there is source-to-source variability within a given source category (e.g. pre-LEV gasoline or DPF-equipped diesel), there are distinct trends in total organic emissions. Gasoline small off road engines (SORE) have the highest emissions, with SORE-2S having, on average, one order of magnitude higher emissions than SORE-4S (Gordon et al., 2013). This is due to less stringent regulations for off-road engine emissions (Cao et al., 2016a), and the unburnt fuel mixing in exhaust due to the two-stoke design in SORE-2S. The LDGV emissions decrease with tightening emission standards (Gentner et al., 2017; May et al., 2014). For example, relative to the median Pre-LEV, there is a 78% reduction in total organic emissions to the median LEV and 90% to the median ULEV. Although not shown here, total organic emission factors are dramatically higher during cold-start than during hot-stabilized operations after the catalytic converter has reached its operating temperature (Saliba et al., 2017). Gas-turbine engine emissions show strong load dependence; idle (4% thrust) emission is comparable to pre-LEV vehicles, and about an order of magnitude higher than high loads (85% thrust) emission. Diesel emissions show strong dependence on both after-treatment devices and test cycle. DPF-equipped diesel vehicles have the lowest emission factors among all tested engine types. Lower emission factors are measured for high speed transient operations (e.g. UDDS cycle) compared to idle/low speed operations. The trends in gas-turbine and diesel emissions are qualitatively consistent with Cross et al. (2013, 2015) who showed similar load-dependent trend of decreasing THC or IVOC emission factors of gas-turbine and diesel engines with higher loads.

As expected, Figure 2 (a) indicates there is source-to-source variation in total organic emission for a given category (e.g. pre-LEV or ULEV). This variability reflects the effects of difference of engine design, engine calibration, after-treatment system, vehicle age, and maintenance history on emissions. However, the previously described trends in total organic emission among the different source categories are clear even with this variability.

## 3.2 Volatility and chemical composition distributions

Figure 3 shows the median volatility distributions of the emissions for three different source categories: gasoline (cold-start), gas-turbine and non-DPF diesel. For gas-turbine engine category, we plot the idle load (4% thrust) emission.

Figure 3 indicates that the organic emissions from all three source categories have tri-modal volatility distributions. The dominant mode is the middle one, with a peak at $C^*=10^8$ µg m$^{-3}$ for gasoline sources, $C^*=10^6$ µg m$^{-3}$ for gas-turbine sources, and $C^*=10^5$ µg m$^{-3}$ for diesel sources. For each source category, this mode has a similar volatility distribution and chemical composition as unburned fuel (Figure S7). We therefore call it the 'fuel mode'.

The fuel mode contributes 72.6% (66.5-77.6% as 10[th] to 90[th] percentile, same hereafter) of the total organic emissions in gasoline engine exhaust, 63.1% (48.9-84.4%) in diesel engine exhaust, and 37.5-38.5% in gas-turbine source emissions. The widely varying contribution of this mode to diesel emissions is due, in part, to after-treatment and duty-cycle effects. For example, low-speed operation (creep and idle) test results show higher mass fractions in the fuel mode, 79.7% (62.2-83.0%), compared to high speed operations. The size of the fuel mode to DPF diesel vehicles is highly variable, 54.2% (29.3-79.9%), which is likely due in part to higher uncertainty associated with measuring very low emission rates.

Figure 3 highlights how the changes in fuel composition create systematic differences in volatility distribution of the emissions among the three source categories. Specifically, the 'fuel' mode of the exhaust shifts towards lower volatility from gasoline to diesel sources mirroring the trend in fuel volatility. Although the chemical composition of the fuel mode is also similar to that of unburned fuel (Figure S7), there are some important differences indicating that combustion and removal efficiencies vary by compound class, which are discussed in section 3.4.

Emissions from each source have a low-volatility mode, comprised of SVOCs and even less-volatile organics. For all three source categories, this low-volatility mode peaks at a $C^* = 10$ µg m$^{-3}$, which is in the middle of the SVOC range. Therefore, some of the organics in the low-volatility mode partition into the particle phase in the atmosphere to form POA, while the rest exist as vapors. The volatility distribution of this mode is similar to that of lubricating oil (May et al., 2013a, 2013b; Worton et al., 2014); we therefore refer to the low-volatility mode as the 'oil mode'. For diesel, the low-volatility and fuel modes blend together. The oil mode contributes 1.4% (0.6-4.2%) of the total organic emissions for gasoline sources, 4.2-12.1% for the gas-turbine source, and 5.9% (3.1-17.7%) for diesel sources.

The size of the LDGV 'oil mode' varies with certification standard, with median values of 0.8% in Pre-LEV, 1.4% in LEV and 2.2% in ULEV. This trend indicates that improvements in after-treatment technology more effectively remove NMOG emission compared to POA emissions. The wide range of SVOC emissions from gas-turbine and diesel sources reflects the effects of changes in engine load/after-treatment: at 85% engine load, 12.1% of gas-turbine emissions are in the 'oil mode' versus only 4.2% at 4% load. DPF-equipped vehicles show 14.8% (10.1-30.6%) of the emissions for on high-speed cycles versus a much lower fraction 2.1% (1.7-5.7%) at low speed operations.

The third mode is the most volatile one, peaking at a $C^* = 10^{10}$ or $10^{11}$ µg m$^{-3}$. It contributes 25.9% (21.1–31.0%) of the total organics in gasoline emissions, 26.9% (9.4-40.6%) in diesel sources emissions, and 50.5-57.3% in gas-turbine engine emissions. It is comprised of the smallest compounds, such as $C_2$-$C_5$ alkanes, alkenes and carbonyls, produced from the incomplete combustion and breakdown of fuel molecules (May et al., 2014). It also contains other compounds such benzene in the $C^* = 10^9$ µg m$^{-3}$. We therefore call it the 'combustion by-product' mode. The composition of this mode varies modestly by source class.

The majority of the IVOC emissions are found in the lower volatility end of the fuel mode. For gasoline sources, IVOCs are in the lowest volatility tail of this mode. For the LDGV tested using cold-start unified cycle, IVOCs contribute 4.5% (2.4%-9.6%) of the total organic emissions. This includes both heavily controlled and low emitting ULEV and less controlled and higher emitting pre-LEVs. IVOCs contribute a similar fraction to the organic emissions from largely uncontrolled and high emitting SOREs (Figure 2). However, IVOCs contribute a larger fraction, 18.1% (5.8 – 31.1%) for organic emissions from LDGV operated over the hot-start unified cycle (Zhao et al., 2016). This suggests that catalytic converters may be less effective at removing lower volatility organics such as IVOCs, which is also consistent with the trends in SVOC data discussed above. However, only four vehicles were tested using the hot-start unified cycle and the IVOC faction varied widely. More research is needed to understand the effects of hot-operations and duty cycle in general on IVOC emissions.

For sources operating on less volatility fuels, IVOCs contribute a larger fraction of the emissions. For example, they contribute 20-27% of gas-turbine engine emissions at idle and 85% loads. This is somewhat larger than data from Cross et al. (2013) who reported 10-20% of NMHC emissions are IVOCs at idle load. The difference could be due to multiple factors, including differences in collection techniques (cryogenic versus adsorbent) and/or differences in fuel composition (Corporan et al.,

2009). Diesel sources emit the highest fraction of IVOCs, with median value of 51.3% (28.7-61.5%). Non-DPF diesel emissions have a more consistent IVOC fraction of 57.1% (46.3-66.4%) than DPF-diesel emissions (40.1%; 17.2-55.5%). Finally, the contribution of IVOCs qualitatively mirror the fuel composition: 1% of unburned gasoline is comprised of IVOCs, ~50% for JP-8, and ~70% for diesel (Corporan et al., 2009; Gentner et al., 2012; May et al., 2014).

Figure 2(c) indicates that the contribution of SVOCs also differs by source type. For gasoline engines, SVOCs contribute 1.1%

(0.4-3.6%) of the total organic emission. This variability is, in part, associated with the effects of tightening emissions certification standards as discussed above. For gas-turbines, SVOCs contribute 3.6-4.6% of total organic emissions. For diesel source, SVOCs contribute 4.6% (2.3-16.1%) of the total organic emission; the wide range reflects effects of duty cycle and after-treatment as discussed above. There are no SVOCs in unburned gasoline and jet fuel, and less than 2% for diesel fuel. The SVOCs in the emissions are likely predominantly from lubricating oil (Worton et al., 2014).

Given that the total organic emissions vary by more than five orders of magnitude (Figure 2a), the volatility distribution (and emissions profiles) are relatively consistent across sources using the same fuel type (Figure 3). As discussed previously, for a given fuel type, after-treatment technology (e.g. LDGV emission certification standard) and test cycle can also influence the volatility distribution, but their influence is much less than that of fuel. We therefore use the median distributions to represent the properties of the aggregate emissions from a large number of sources with a given source category. There is always source-

to-source variability, but for inventories we need to define representative profiles for distinct categories (we use medians as opposed to averages to reduce the influence of outliers).

An important question is the number of distinct source categories. To investigate this question, Figure 4 compares the volatility distributions of different sources in a 2-dimensional space of IVOC versus SVOC mass fractions. These are important SOA precursors so this framework highlights differences in SOA formation potential. There are three distinct clusters in Figure 4,

one for each fuel type (gasoline, diesel and jet). Therefore, these source categories require different profiles. For example, the on-road (cold-start) and off-road gasoline sources emissions cluster, with a median mass IVOC and SVOC fractions of 4.5% and 1.1%, respectively, indicating similar volatility distributions between on- and off-road gasoline sources. Figure 4 also suggests two additional categories, but these distinctions are not as strong given the variability of the data. First, hot-start LDGV emission have much higher IVOC and SVOC fractions than cold-start emissions (18.1% versus 4.5%, 4.7% versus

1.1%). This implies a roughly 4-time higher SOA yield for hot-start on-road gasoline emissions. Therefore, separate profiles should be used to represent cold-start and hot-operation emissions when constructing emission inventories for gasoline vehicles. Second, DPF and non-DPF equipped diesel sources also show significant different volatility distributions, especially in SVOC mass fraction (12.2% versus 3.8%). To account for these differences, we present five emission profiles in Table S3: gasoline (cold-start and off-road), gasoline (hot-start), diesel (non-DPF), diesel (DPF) and gas-turbine engines. Interestingly,

the SVOC and IVOC mass fractions are strongly positive-correlated across all sources with an exponential fit between SVOC and IVOC mass fraction of $f_{SVOC} = 0.100 \, f_{IVOC}^{\,0.700}$. One could certainly define additional source categories to, for example, account for trends in SVOC fraction with emission certification of LDGVs, but it is not clear that those difference are large enough to improve model performance versus using an aggregate profile to represent all gasoline vehicles.

## 3.3 New versus traditional source profiles

Figure 3 also compares our new comprehensive source profiles to traditional profiles used to construct the emission inventory to simulate air quality in the Los Angeles region during the 2010 CALNEX campaign (Baker et al., 2015). Our new profiles are the median value of the measured emission for gasoline (separate for cold-start and hot-operations), gas turbine, non-DPF and DPF-diesel sources; they are listed in Table S3 (Supporting Information). The traditional profiles are from the EPA SPECIATE database (USEPA, 2016b) -- profile #4674 for diesel, #8750a for gasoline, and #5565 for gas turbine sources with the POA fraction (NVOC) calculated using MOVES (USEPA, 2014) .

There is good agreement between our new and traditional profiles in the VOC range, with both having similar chemical compositions and volatility distributions containing both by-product and fuel modes (Figure 3). For example, Figure 5 demonstrates the strong agreement for SARPC-lumped VOC groups between the new and traditional profiles for all three source categories with more than 90% of the SAPRC groups for the gasoline sources agree within a factor of two. We recommend using our new profiles for VOC composition because they have enhanced VOC speciation from combining the existing SPECIATE profiles with our new experimental data.

However, the traditional profiles dramatically underestimate IVOCs and SVOCs, which are important classes of SOA precursors. As illustrated in Figure 1, this is a consequence of the limitations of traditional source characterization techniques to quantitatively collect and analyze IVOCs. For example, the traditional LDGV emission profile only attributes 0.2% of the total organics to IVOCs versus 4.6% in our new cold-start profile. The traditional gas-turbine engine emission profile attributes 13% of the organics to IVOCs versus 27% IVOCs in our new profile. For diesels vehicle emissions, the traditional profile attributes 10% of total organic emission to IVOCs versus 54.2% of organics for non-DPF diesel in our new profile. The traditional diesel source profile does contain about 20% unknown organics (UNK), part of which are likely IVOCs, as the collection and chemical analysis efficiency decrease towards lower volatility bins such as $10^3$ and $10^4$ µg/m$^3$ (Figure 3). However, most UNK is not represented as IVOCs in models, as discussed in section 2.2.

## 3.4 Exhaust versus unburned fuel and IVOC enrichment factors

Figure 3 highlights the large contribution of unburned fuel to the exhaust, but careful examination of the data reveals that the combustion process and/or removal efficiency by the after-treatment device are compound dependent. For example, gasoline and gas-turbine emission are both enriched in IVOCs compared to fuel (e.g. C*= $10^6$ µg/m$^3$ for gasoline, and C*= $10^4$ µg/m$^3$ for gas-turbine). The difference suggests higher combustion efficiency of more volatile fuel components.

Figure S8 compares the chemical composition of the exhaust to unburned fuel. Overall, straight and branched alkanes (speciated and unspeciated) contribute a smaller fraction to the exhaust than in the fuel with the median mass fractions decreasing from 46.6% (fuel) to 34.3% (exhaust) for gasoline sources, 50.0% to 9.8% for gas-turbine source, and 30.3% to 11.2% for diesel sources. In comparison the fraction of aromatic and cyclic compounds (speciated and unspeciated) are consistent between fuel and exhaust; for example, 37.2% (exhaust) versus 36.1% (exhaust) for gasoline source and 58.7% to 60.2% for diesel source. This comparison implies higher combustion efficiencies of $n$-/$b$- alkanes than cyclic/aromatic compounds in internal combustion engines, which could partly be explained by the flash points of different hydrocarbons. The mass fraction of alkenes, alkynes and carbonyls increase, indicating they are important product of incomplete combustion. For example, they increase from 3.5% (fuel) to 28.6% (exhaust) for gasoline sources, and 0% to 54.5% and 24% for gas-turbine and diesel sources, respectively. Gasoline emission have the highest single-ring aromatics fraction (~30%), compared to 5.5% in gas-turbine and 17% in diesel emissions. This mirrors fuel composition -- unburned gasoline fuel had the highest aromatic content (26.7%) of the fuels tested here.

We are especially interested in the enrichment or depletion of SOA precursors in the exhaust compared to fuel, including IVOCs and single ring aromatics. To quantify enrichment, we normalized SOA precursors in both the fuel and exhaust to $C_8$ to $C_{10}$ $n$-alkanes, a tracer for the unburned fuel. As shown in Figure S7 and S9, some SOA precursors are enriched, and others depleted relative to fuel. Benzene and total IVOCs in gasoline and toluene and $C_8$ aromatics in diesel exhaust are enriched by more than a factor of two relative to unburned fuel. Enrichment of single-ring aromatics are likely due to pyrolysis of larger aromatic molecules (Akihama et al., 2002; Brezinsky, 1986). In contrast, total IVOCs (normalized to the $C_8$ to $C_{10}$ $n$-alkanes) are depleted in diesel exhaust compared to fuel (enrichment factors less than 1).

Figure 6 shows box-whisker plots of the total IVOC enrichment factors. Sources using more volatile fuel have higher IVOC enrichment factors. For example, relative to $C_{8-10}$ $n$-alkanes, gasoline engine exhaust has a median total IVOC enrichment factor of 8.5 versus modest depletion (enrichment factor <1) in diesel source exhaust with gas turbine exhaust in between. There are several possible explanations for this trend. IVOCs may be less efficiently combusted in the engines. Recent research also shows that less IVOCs are removed by catalytic converters compared to VOCs (Pereira et al., 2017). Figure S10 plot the IVOC enrichment factors of Pre-LEV, LEV and ULEV vehicles exhaust. Due to the different removal efficiency between IVOCs and VOCs, median ULEV vehicles show even higher (>10) IVOC enrichment factor. Lubricating oil decomposition products may also contribute to the IVOC emissions (May et al., 2013a; Worton et al., 2014). Finally, the IVOC fraction in fuel may be underestimated due to limitations in techniques used commonly to characterize fuel composition (Gentner et al., 2012).

**4 Implications for OA formation**

An important goal of this work is to develop emission profiles required to improve model predictions of SOA formation. Simulation of ambient OA concentrations requires accurate representation of both emissions and SOA yields for SVOCs and IVOCs. Given the lack of IVOC data in traditional source profiles (Figure 3), previous modelling studies have used different

scaling approaches, most commonly based on POA (Koo et al., 2014; Murphy et al., 2017; Robinson et al., 2007; Woody et al., 2016) but also using NMOG (Jathar et al., 2014, 2017) and naphthalene (Pye and Seinfeld, 2010). Finally, Gentner et al. (2012) used unburnt fuel surrogate to estimate IVOC emissions. These estimates are then combined with SOA yield data.

In this section, we use our new data to evaluate these different scaling approaches for estimating IVOC emissions to better understand their strengths and limitations for simulating ambient OA concentrations. Table 1 lists the different approaches we evaluated: (1) New – new profiles developed in this paper; (2) Trad – traditional profiles (SPECIATE #8750a for gasoline, #5565 for gas-turbine and #4674 for diesel sources emissions); (3) ROB: traditional profiles + 1.5 × POA as IVOCs (Robinson et al., 2007); (4) MUR: traditional profiles + 9.656 × POA as IVOCs (Murphy et al., 2017); (5) PYE: traditional profiles + 66 × Naphthalene as IVOCs (Pye and Seinfeld, 2010); (6) GEN: using unburnt fuel composition as surrogate (Gentner et al., 2012); (7) JAT: 20% of NMOG of gasoline emission and 25% of diesel emissions are IVOCs (Jathar et al., 2014).

Figure 7 compares our new data to the six previous estimates. Figure 7(a) shows the mass fraction of different classes of SOA precursors (VOC, IVOC and SVOC) in the NMOG emissions. Figure 7(b) shows the overall SOA yields of the total NMOG emissions for the different models (SOA mass/mass of NMOG emissions).

As shown in Figure 7(a), all estimates have similar VOC SOA precursor mass fractions, but widely divergent amounts of IVOCs. Our new profiles (1) and estimates (6) and (7) have modestly lower VOC SOA precursors, due to the inclusion of IVOCs and gas-phase SVOC within NMOG emissions, while approaches (3) – (5) add additional IVOCs to the existing NMOG emissions. Since FID-based NMOG is a measure of all non-methane organic gases, we think IVOCs and gas-phase SVOCs are largely accounted for in existing NMOG emission factors for the types of sources measured here (see discussion of mass closure between bulk measurements and speciated measurements in section 2.2 and Figure S3). However, traditional source profiles do not correctly attribute these emissions to IVOCs/SVOCs in all sources.

The most common approach to incorporate IVOCs in models has been to scale POA emissions as defined by the organic mass collected on a quartz filter. The scaling ratios (e.g. IVOC-to-POA) were estimated from very limited data (a single or small number of sources) and the same ratio has typically been applied to all source categories. Our data indicate that scaling with POA is not a robust approach because IVOC-to-POA ratios vary by source category. For example, the average IVOC-to-POA ratios for gasoline engines exhaust are 6.2 ± 4.4 (cold-start) versus 12 ± 7 (non-DPF equipped) and 31 (DPF-equipped) for diesel exhaust (Zhao et al., 2015, 2016). In addition, these values are much larger than the widely used scaling factor of IVOC-to-POA of 1.5 (ROB in Figure 7) (Robinson et al., 2007), which grossly underestimates the IVOC emissions from the types of internal combustion engines considered here. While the IVOC-to-POA ratio of 9.6 by Murphy et al. (2017) (MUR in Figure 7) overestimates IVOC emissions from gasoline and gas-turbine sources, but underestimates it from diesel sources.

However, even if one uses source specific IVOC-to-POA scaling factors, we do not think that scaling POA provides a robust estimate IVOC emissions from internal combustion engines. POA emissions are dominated by lubricant oil (Worton et al., 2014) while IVOC emissions appear to mainly arise from unburned fuel (Figure 2). In addition, quartz filter measurements are subject to sampling artifacts and partitioning biases (May et al., 2013a, 2013b, 2013c). As a result, IVOC-to-POA ratios vary not only by source type (e.g. gasoline versus diesel) but also operating conditions (Zhao et al., 2015).

Zhao et al. (2015, 2016) reported stronger correlations between IVOC and total NMOG emissions than with POA over a range of operating conditions ($R^2$ = 0.96 vs 0.90 for gasoline and $R^2$ = 0.99 vs 0.91 for diesel sources, Figure S11). This is not surprising given that both NMOG and IVOC emissions arise from fuel and are controlled by similar processes. This suggests that IVOC emissions should be estimated using source specific scaling factors of NMOG not POA.

Jathar et al. (2014, 2017) estimated IVOC emissions by scaling NMOG. They also used different ratios for gasoline and diesel sources. However, they did not directly measure IVOCs. Instead they inferred IVOC-to-NMOG ratios using a combination of unspeciated emissions and inverse modelling of SOA production measured in a smog chamber. Using this approach, they attributed 25% of NMOG emission from gasoline engine and 20% from diesel engines to IVOCs. These values are very different than those reported here, which are based on direct measurements. A detail on the ratios of Jathar et al. (2014) is that

they were derived to be used in combination with their empirically derived SOA yields. When used together they explain SOA yield production measured in smog chamber experiments with dilute exhaust. Therefore, one cannot simply replace IVOC-to-NMOG of Jathar et al. (2014) with the ones reported here without also using different SOA yields.

Pye and Seinfeld (2010) estimated IVOC emissions by scaling naphthalene using the same IVOC-to-naphthalene ratio for all sources. Our data indicate that naphthalene is not a good indicator of IVOCs, due to the large variation in fuel aromatic content.

For example, there is four times more naphthalene in gasoline engine exhaust (0.4%) and fuel (0.13%) compared to diesel engine exhaust (0.1%) and fuel (0.04%). Therefore, the approach of Pye and Seinfeld (2010) generates much higher estimates of IVOC emissions from gasoline than diesel sources, which is opposite of the actual emissions data (Figure 2). In principle, this problem can be overcome with source specific IVOC-to-naphthalene ratios, but even with source-specific ratios, individual organics are likely a less robust scaler for IVOCs than total NMOG because fuel composition (e.g. aromatic content) varies by

location and season.

A final approach to estimate IVOC emissions is to use unburned fuel as a surrogate for the SOA production of exhaust. Gentner et al. (2012) used this approach to estimate the IVOC fraction, as well as the SOA yield of gasoline and diesel engine exhaust. This approach works for diesel, but not for gasoline given the enrichment of IVOCs in the exhaust (Figure 6).

In Figure 7(b), we combine the different emissions estimates with SOA yield data to calculate the SOA yield of the NMOG

emissions for each source category, assuming complete oxidation of all precursors. Our new profiles predict that IVOCs and SVOC vapors contribute substantially to SOA production, especially for sources using lower-volatility fuels (e.g. diesel). For gasoline sources, we predict that IVOCs and SVOCs contribute as much SOA as traditional VOC precursors (mainly single-ring aromatics). Accounting for IVOCs in gasoline exhaust almost doubles the predicted SOA production compared to the traditional profile. For gas-turbine and diesel sources, IVOCs and SVOC vapors combining contribute factors of 13 and 44

more SOA than VOCs, respectively.

Figure 7(b) also compares the SOA yields of NMOG emissions for all the different approaches (2) – (7). The differences in effective yields are primarily due to differences in IVOC/SVOC emissions. Traditional profiles and ROB underpredicts SOA production from all three source categories because they underestimate IVOC emissions. As discussed in section 3.4, IVOCs are enriched in gasoline emissions compared to unburned fuel, therefore GEN underpredicts the SOA yield of gasoline

emissions. However, fuel composition provides a reasonable estimate for SOA production from diesel emissions, except for the lack of SVOCs potentially produced from the usage of lubricant oil. The approaches of PYE and JAT overpredict the overall SOA production from gasoline emissions, due to their overestimation of IVOC emissions, but both underestimate the overall SOA production for diesel emissions.

To conclude, none of the previous modelling approaches provide a robust estimate of the IVOC fraction in the exhaust for all three source categories. Figure 7(a) and (b) show that traditional profiles either completely omit IVOCs or incorrectly lumped them to VOC chemical mechanism groups, which greatly underestimate the overall SOA formation potential. Approaches (3) – (5) apply scaling factors to certain species, such as POA and naphthalene, but these factors vary by source and fuel composition, which may lead to significant bias for different sources. Using unburnt fuel composition as a surrogate in

estimation (6) only works for sources that use lower volatility fuel, such as diesel.

In addition to better representing gas-phase SOA precursor emissions, the new profiles also account for the semi-volatile character of POA. Partitioning calculations predict that 40% to 50% of traditionally defined POA mass evaporates at typical atmospheric conditions (T=298K and OA=10 µg/m) (May et al., 2013b, 2013a).

## 5 Recommendations and future research needs

Figure 7 highlights the importance of including IVOC and SVOC emissions in models and inventories to improve predictions of SOA formation. This paper facilities this by providing model-ready profiles that include direct measurements of IVOCs and SVOCs. They are designed to be applied to existing inventories of POA and NMOG emissions. These profiles (Table S3) are normalized to total organic emissions (VOC, IVOC, SVOC, LVOC plus NVOC), and therefore should be applied to the sum of gas- and particle-phase organic emissions. Since current emission inventories report gas- (NMOG or VOC) and particle-

phase (PM or POA) emissions separately, the comprehensive profile can be separated into two parts: gas-phase (VOC and IVOC) and particle-phase (SVOC and less volatile components) profiles. These sub-profiles would be renormalized and then applied to the existing NMOG or POA emissions. With this approach, one needs to correct the POA data for missing SVOC vapors not collected during vehicle certification testing (the factor of 1.4 for LDGV discussed at the beginning of section 3).

Our new profiles intentionally do not define the phase state of the emissions. Phase state is not a property of the emissions, but

determined by the combination of the volatility distribution of the emissions and atmospheric conditions because gas-particle partitioning depends on the concentration of organic aerosol and temperature. The profiles specify the volatility distribution of the emissions, which can then be used to calculate the gas-particle partitioning (phase state) for any atmospheric condition (Robinson et al., 2010). This approach is critical to correctly predict POA concentrations for sources that have substantial SVOC emissions, such as the sources tested here and biomass smoke (May et al., 2013c).

The three types of sources considered here account for 98.2% of the mobile source emissions in the 2014 US EPA National Emission Inventory. For other liquid-fuel internal combustion engine sources, we recommend interpolating based on fuel composition and applying the IVOC enrichment factor estimated from fuel volatility (Figure 6). For sources profiles that only

contain speciated VOCs and unknown residual, we recommend not normalizing to known species, as this will likely misattribute low volatility organics to VOCs.

The emission profiles described here (except for gas turbines) are based on experiments conducted using sources recruited from the California in-use fleet, at typical California summertime temperatures (10-25 °C) and using California commercial summertime fuels. Therefore, the data are most directly relevant to California summertime conditions. Ambient temperature can have a large influence on emissions. For example, George et al. (2015) measured about 10 times higher non-methane hydrocarbon (NMHC) emission rates during testing at -7 °C versus 24 °C. The VOC composition also changed with temperature with the fraction of $C_{9+}$ aromatics almost doubling at low temperature. These data suggest that winter emissions may have higher content of larger aromatics ($C_{9+}$ aromatics) and IVOCs, due to incomplete combustion or lower efficiency of catalytic converter. Our profiles therefore likely represent lower bounds to winter vehicle emission in terms of aromatics and IVOC contents. As discussed in section 3.2, unburned fuel is an important contributor to the emissions. Therefore, variations in fuel composition by, for example, season and/or location will influence the composition of the emissions. From an SOA formation perspective, we are most interested in changes in fuel IVOC and aromatic content. Figure S12 compares our new VOC profiles with data from China (Cao et al., 2016; Yao et al., 2015). There is good agreement for many compounds, but not all.

Future research needs:

1) IVOC and SVOC emissions data from vehicles operated over a wider range of conditions. Comprehensive emissions data are needed for a wider range of fuel compositions, test cycles (hot operations), and seasons (especially winter). However, given their major contribution to SOA formation, we recommend using our new profiles even for studies outside of California if the only other option is to use traditional profiles that don't include IVOC and SVOC data.

2) IVOC and SVOC emissions data for non-mobile sources. Recent research has demonstrated that IVOCs and SVOCs are important contributors to biomass burning, oil sands, oil production, and volatile chemical product emissions (de Gouw et al., 2011; Hatch et al., 2017; Hunter et al., 2017; Liggio et al., 2016). More comprehensive, model-ready profiles that account for the full spectrum of organic emissions are needed for these and other source categories (McDonald et al., 2018).

3) Inclusion of IVOCs in air quality models and inventories. Our new profiles are designed to directly incorporate IVOCs into models and inventory. Since they are based on direct measurements, they do not have the large uncertainties associated with the previously developed scaling approaches.

4) Improved chemical composition of IVOCs and SVOCs. Although we have quantified the total IVOC emissions, the majority of these emissions were not resolved at the molecular level. Since the SOA yield of compounds depend on both molecular structure and volatility (Tkacik et al., 2012; Ziemann, 2011), future studies are needed to more fully speciate IVOCs and SVOCs in order to identify the class of compounds that needed for photo-oxidation experiments (Chan et al., 2013; Cross et al., 2015; Isaacman et al., 2012b).

5) Measurements and source apportionment of atmospheric IVOCs / SVOCs. Ambient measurements of IVOCs / SVOCs are needed to identify other important sources of atmospheric IVOCs / SVOCs. This will help future studies to prioritize which sources to characterize.

**Author contribution**

5  Q.L., Y.Z. and A.L.R. designed the research. Q.L. and Y.Z. analyzed the data. Q.L., Y.Z. and A.L.R. wrote the paper.

**Competing interests**

The authors declare no conflict of interest.

**Acknowledgments**

This publication was developed as part of the Center for Air, Climate, and Energy Solutions (CACES), which is supported
10  under Assistance Agreement No. RD83587301 awarded by the U.S. Environmental Protection Agency. It has not been formally reviewed by EPA. The views expressed in this document are solely those of authors and do not necessarily reflect those of the Agency. EPA does not endorse any products or commercial services mentioned in this publication. The authors would like to thank Dr. Albert Presto, Dr. Timothy Gordon and Dr. Andrew May for their help in providing detailed datasets information. The authors would also like to thank Dr. Benjamin Murphy and Dr. Havala Pye for helpful comments on this
15  manuscript and valuable discussion on missing SVOC scaling approach.

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

**Tables**

Table 1 List of different estimates of IVOC emissions and SOA yield for mobile sources shown in Figure 7.

| Label | IVOC Emissions | IVOC SOA yield (at OA = 10 μg/m$^3$) | Reference |
|-------|----------------|--------------------------------------|-----------|
| **New** | Direct measurements | 0.22 - 0.30 | This work, Zhao et al. (2015, 2016) |
| **Trad** | N/A | N/A | EPA SPECIATE |
| **ROB** | 1.5 × POA | 0.15[a] | Robinson et al. (2007), Koo et al. (2014) |
| **MUR** | 9.656 × POA | N/A | Murphy et al. (2017) |
| **PYE** | 66 × Naphthalene | 0.22 | Pye and Seinfeld (2010) |
| **GEN** | Unburned fuel (1% of NMOG for gasoline, 62% for diesel) | 0.034 - 0.20 | Gentner et al. (2012) |
| **JAT** | Inverting chamber measurements (25% of NMOG for gasoline, 20% of NMOG for diesel) | 0.22 - 0.35 | Jathar et al. (2014) |

[a] From Koo et al. (2014)

**Figures**

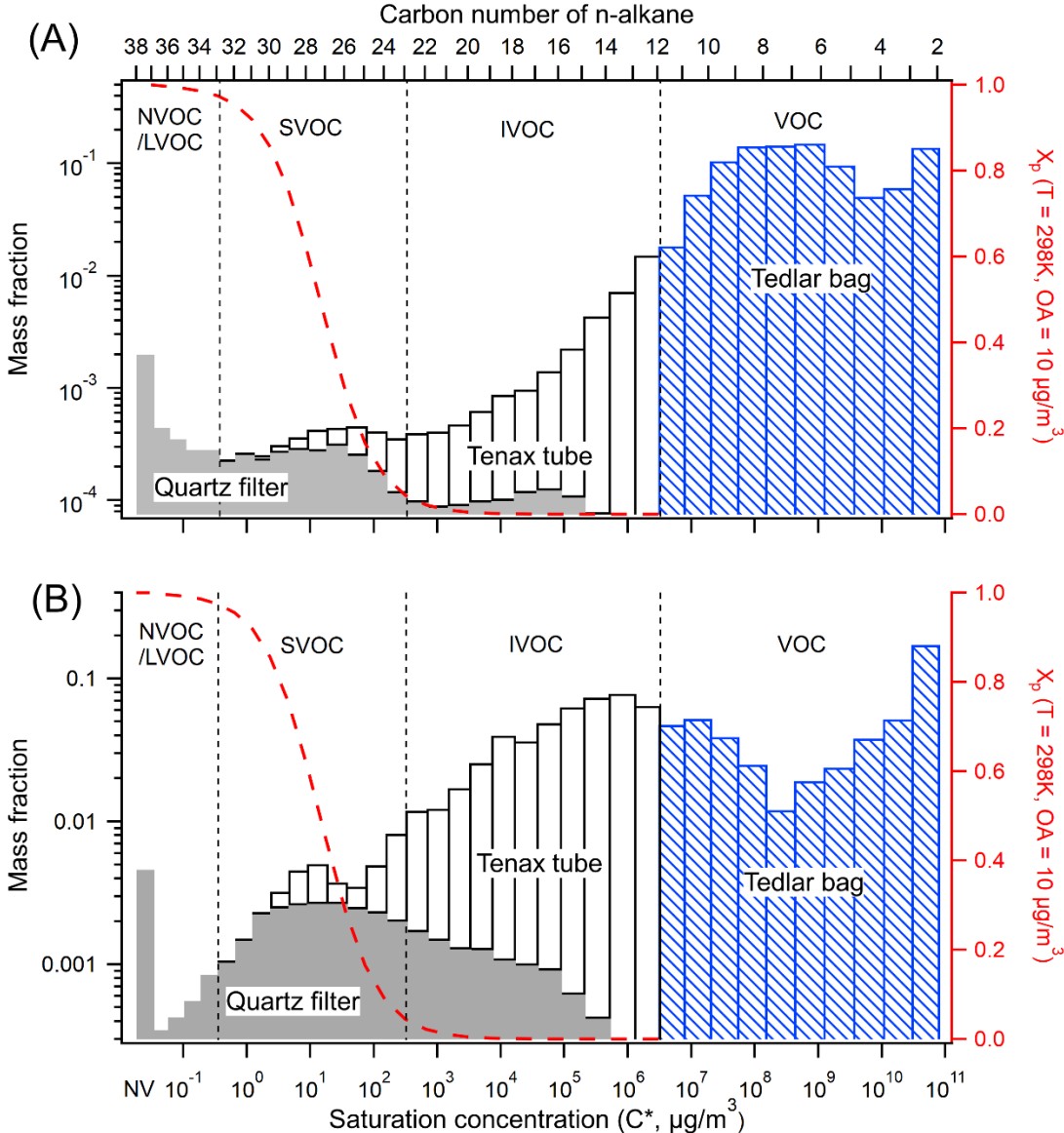

**Figure 1** Volatility distribution of organic emissions for a typical (a) gasoline (b) diesel vehicle. The emissions are classified by sampling media (line 1: Tedlar bag, line 2: bare quartz filter followed by two Tenax tubes). The red dashed line indicates the particle fraction assuming the emissions form a quasi-ideal solution at a $C_{OA}$ of 10 μg m$^{-3}$ and temperature of 298K.

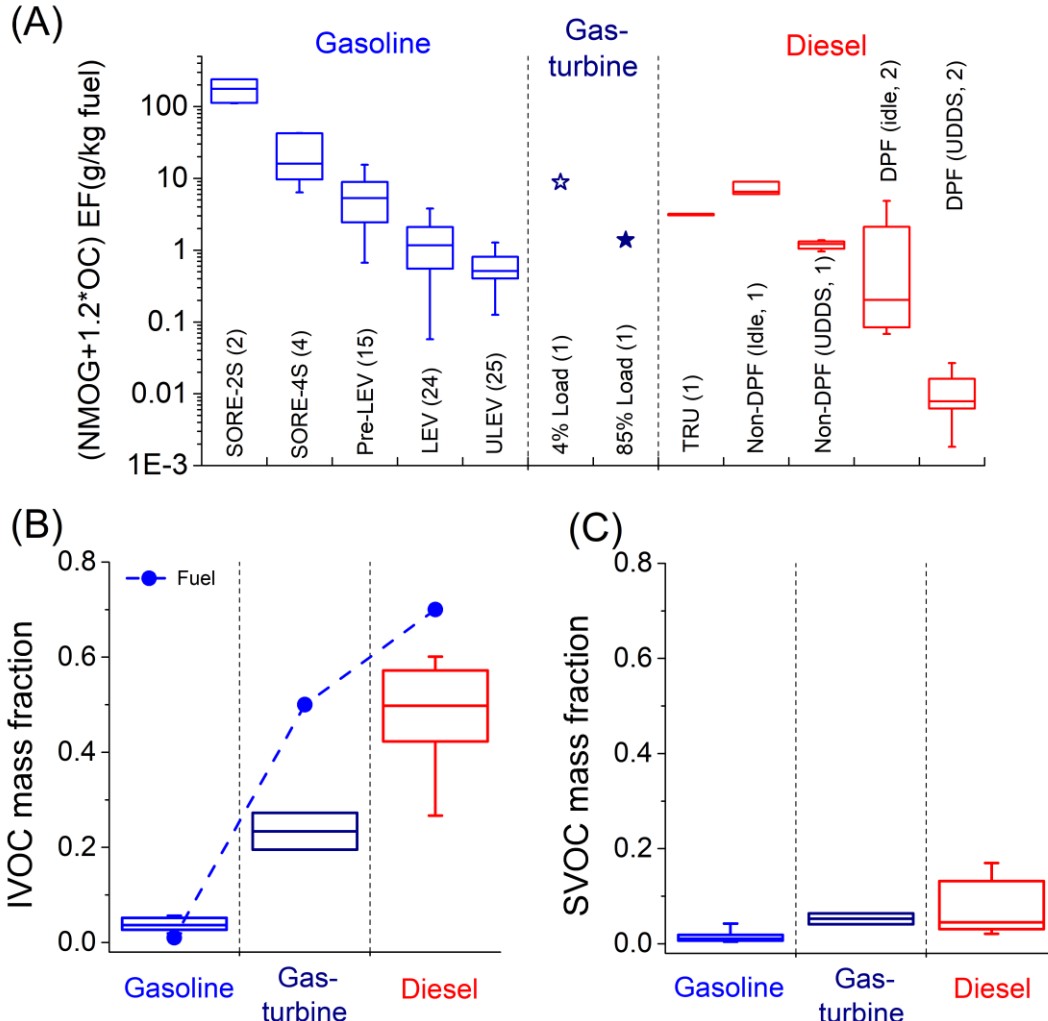

**Figure 2 (a) Emission factors for total organics (NMOG+1.2*OC) for different source categories. The number in parentheses indicates number of unique sources tested in each category. Mass fraction of (b) IVOCs and (c) SVOCs in total organics for each source category. This figure only shows cold-start and off-road gasoline engine emissions. Box-whisker plot represents range of emission for each category: 25th -75th percentiles and 10th-90th percentiles.**

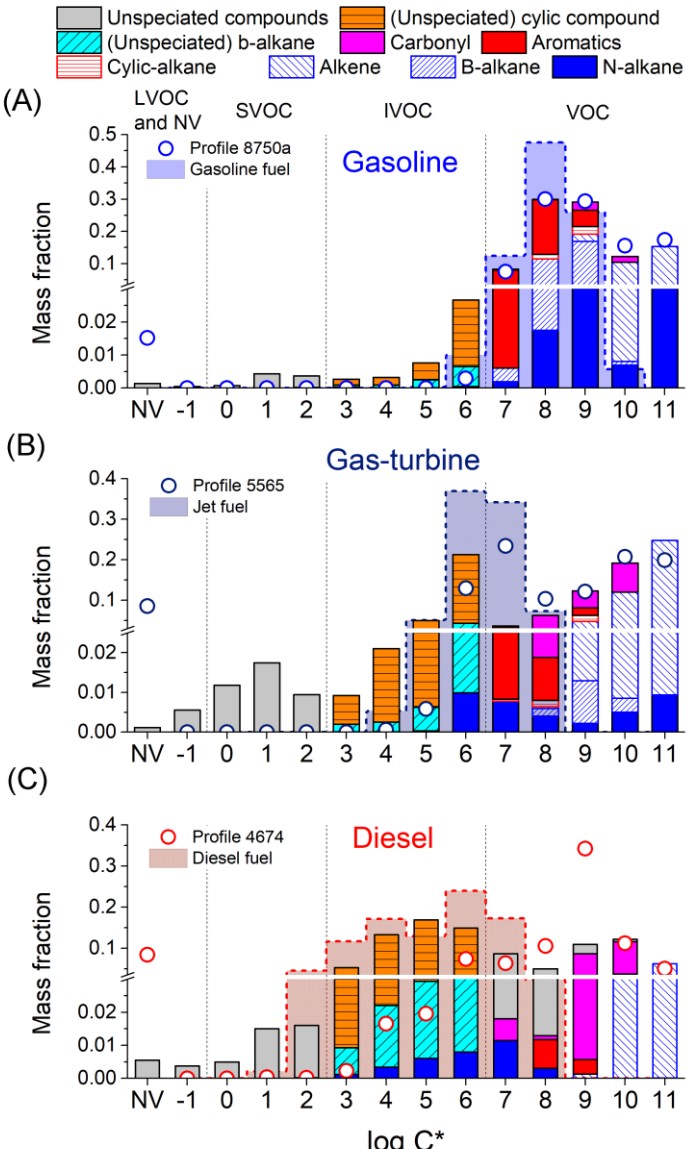

**Figure 3 Median volatility distribution of organic emissions for (a) cold-start on-road gasoline, (b) gas-turbine (idle) and (c) on-road non-DPF diesel engines. The color shading indicates composition. Shaded area indicated by dashed indicate distribution for unburned fuel; dots indicate traditional source profiles from EPA SPECIATE database. The y-axis has a broken scale to amplify the least volatile emissions.**

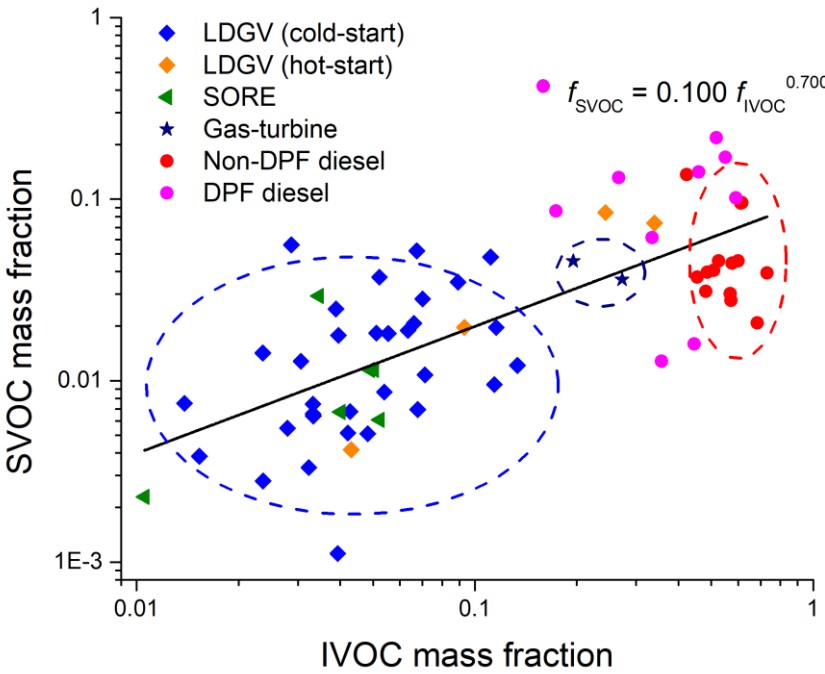

**Figure 4 Two-dimensional visualization of volatility distributions (x axis: IVOC mass fraction, y axis: SVOC mass fraction) of all tested sources. Dashed circles indicate clusters by fuel type: Blue cluster: gasoline (cold-start), navy: gas-turbine, red: non-DPF diesel source.**

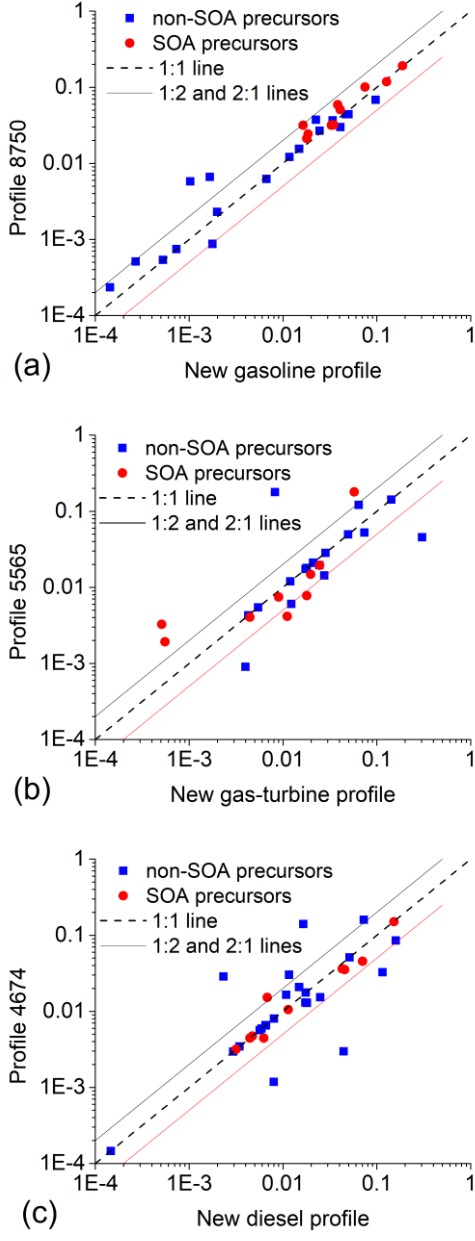

**Figure 5 Scatter plot of VOC groups in SAPRC mechanism in the new profiles and in SPECIATE database for (a) on-road gasoline, cold-start (b) gas-turbine (c) non-DPF diesel sources, demonstrating consistency between traditional and new profiles in VOC speciation**

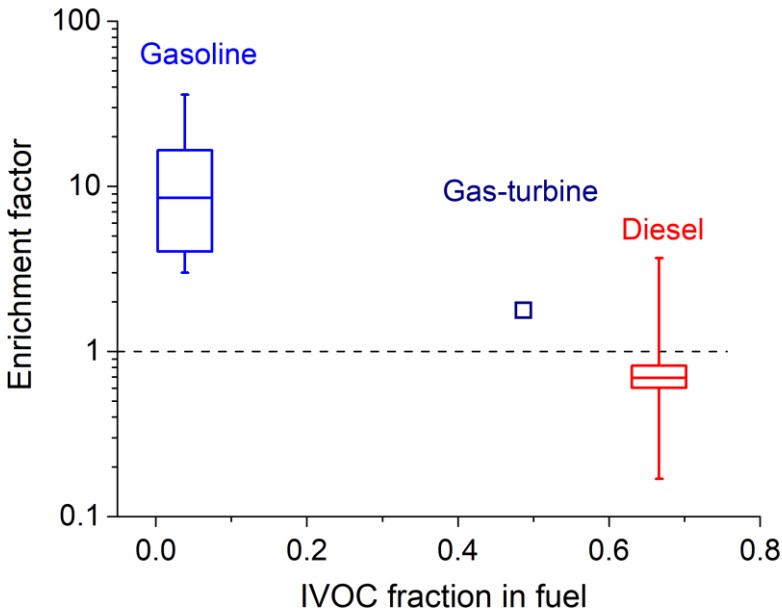

**Figure 6 IVOC mass enrichment factors as a function of IVOC content in fuel, $R_{Enrichment,i} = (m_i^{exhaust}/m_{C8-10}^{exhaust})/(m_i^{fuel}/m_{C8-10}^{fuel})$. The box-whisker plots indicate variability in ratio within a given source class: 25th -75th percentiles and 10th-90th percentiles.**

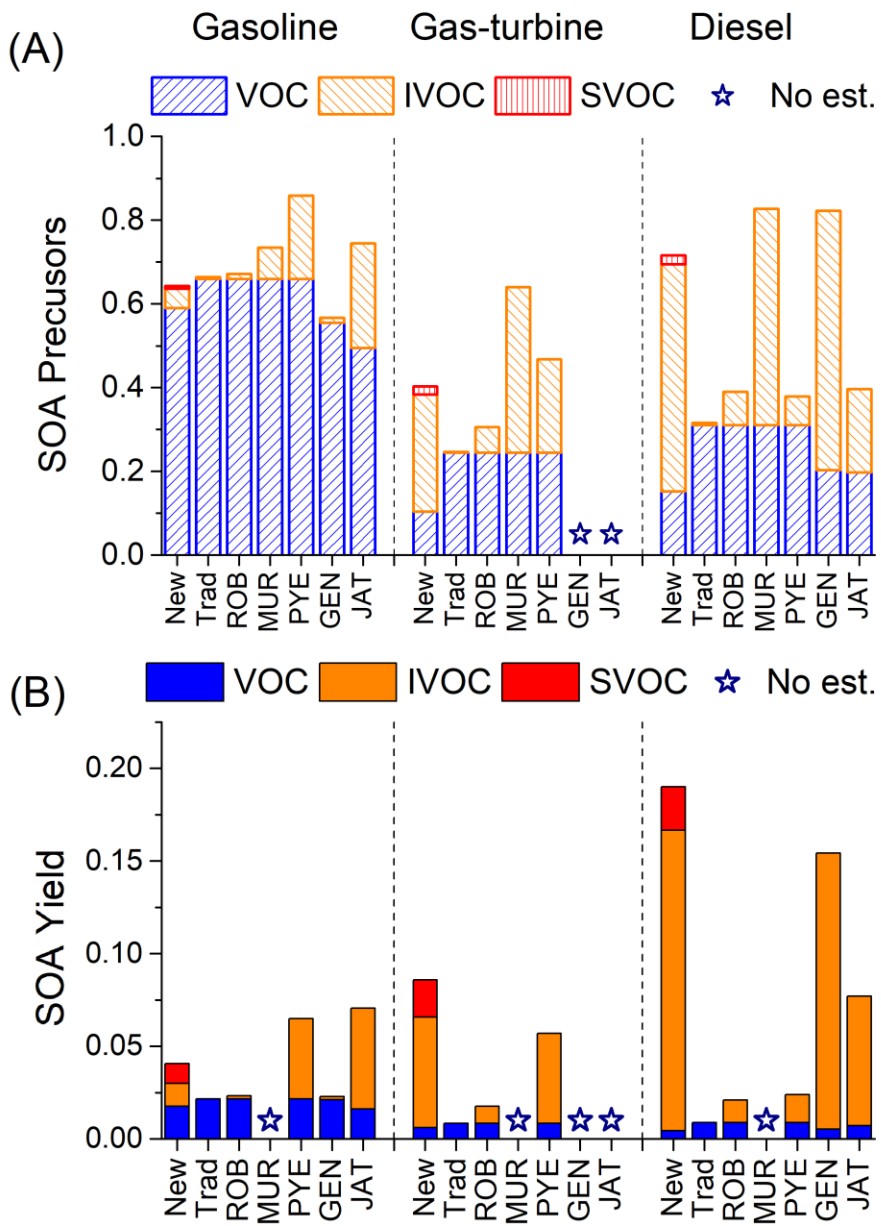

**Figure 7** Comparison of (a) mass fraction of SOA precursors in total NMOG emissions and (b) calculated total SOA yields of NMOG emissions from mobile sources based on the different estimation approaches listed in Table 1. Star denotes no estimate available.