# Peer review of "Comprehensive organic emission profiles for gasoline, diesel, and gasturbine engines including intermediate and semi-volatile organic compound emissions"

_Atmospheric Chemistry and Physics, 2018_

## Referee Comment (RC1) · Anonymous Referee #1 · 21 Aug 2018

**Comments on the "Comprehensive organic emission profiles for gasoline, diesel, and gas-turbine engines including intermediate and semi-volatile organic compound emissions" by Quanyang Lu et al.**

*General comments:*

The paper compiled a comprehensive model-ready organic emission profiles covering four important mobile sources: on-road and off-road gasoline, gasoline-turbine, and diesel engines. Mass fractions of VOCs (volatile organic compound), IVOCs (intermediate-volatile organic compounds) and SVOCs (semi-volatile organic compounds) were determined and analyzed systematically. This work pointed out that proportions of IVOCs and SVOCs are relatively consistent with the fuel type. The fractions of IVOCs for diesel engines can be as high as 51.3% in the total organic mass. This work demonstrated the importance of quantifying the mass of IVOCs and SVOCs using the "correct" techniques in atmospheric models for SOA prediction, which provides important insights for modelers, inventory development and profile measurement.

The whole manuscript is well organized and clearly written. Data, figures and tables can generally support the argument. The main concern is on the unclear illustration of the phase state for the total organic matter, i.e., which part of the SVOCs and IVOCs are emitted at the condensed phase and can be classified as POA, and other parts as NMOG at a typical atmospheric condition? A linkage between the VBS volatility bins and the phase state in real world should be set up. Since the SPECIATE profiles (#4674 and #5565, #8750) were measured only for mass distributions of the gases, comparing these profiles with the "organic emission" profiles are not reasonable. Technically, the author is suggested to use the conjunction words more carefully to logically organize the sentences.

I recommend the manuscript to be revised considering the following comments.

*Specific comments:*

*Abstract:*

1. Line 18-19: Relatively large variances of the mass fractions for SVOCs are found for the diesel engines (3.1%-17.7%), even using the same fuel type, which means that the end-of-pipe control at least for the diesel engines can effect the mass distribution and cannot be neglected. The sentence of "this suggests that a single profile can be used to represent the emissions from sources operating on the same fuel" is not rigorous and cannot be supported by the main text. Please verify the sentence or specify the conditions.

*Introduction:*

2. Page 2: The abbreviations of IVOC, SVOC and LVOC should be explained when they appear in the main text for the first time. Please add them.

*Methods:*

3. Since there are several procedures in compilation of the profiles, mapping organics into the VBS, SOA evaluation and also technical details in this section, a diagram illustrating the whole processes would be useful to outline the method more clearly.

4. Page 3, line 14: You mentioned that "slightly different procedures were used to characterize VOC emissions" confuse me. Can you specify it more clearly? One summary sentence to describe the differences and how much effect on the VOC emissions will help the readers to follow.

5. Page 4, line 31: As my understanding, the SAPRC mechanism or the CB mechanism are only applied for the gas-phase VOC reactions in atmospheric models. For the organic emissions in the particle phase, they are treated as OM/POA. Please clarify the statement.

6. Page 5, line 10: Is $C^*$ varying with the temperature? The calculated $C^*$ in this paper is for T=298K? With the dilution of the vehicle exhaust, the temperature is supposed to decrease gradually to the ambient temperature, to what degree will the $C^*$ change during this process? Can you add some illustrations on the variations of $C^*$ and phase state of the organic compound with temperature, maybe in the supplement?

7. Page 5, line 15: "represents" should be "represent".

8. Page 5, line 27: Please add one or two summary sentences describing the comparison results and conclusions for Figure S1 (a-c).

9. Page 5, line 11: Please explain the abbreviation of "LDGV" when it first appears.

*Results and discussion:*

10. Page 7, line 9: Could you provide a table listing the calculated SOA mass yields for IVOCs in the supplement?

11. Page 7, line 24: You mentioned that there is also substantial breakthrough of SVOCs from the quartz filter indicated from Figure 1. I may miss some important features in Figure 1. Can you explain more on this "breakthrough" based on Figure 1?

12. Page 9, line 3: The author is suggested to be more careful to use the conjunction words. Since you don't discuss about the phase state for SVOCs or the partition status between gas phase and particle phase for SVOCs in the context, the statement of "the peak in this low-volatility mode is in the middle of the SVOC range" cannot support the following conclusion of "some of the organics are in the particle phase and some of them in the gas phase". More discussions on the partition between gasparticle phases for IVOCs/SVOCs are needed.

13. Page 9, line 14: Aromatics is also important for this mode, especially for diesel engines.

14. Page 10, line 13: As my understanding, the traditional profiles are only for the gas-phase VOCs, which were measured using Tedlarbag / canister followed by GC-MS for hydrocarbons, and 2,4-DNPH followed by HPLC for carbonyls. OC is measured and corresponding emissions is assigned in the particle phase. It is not reasonable to compare the mass distributions between "total organic emission" and "gas-phase VOC". Only gas-phase emissions should be included in the comparison, or OC should be also included in the traditional profile to represent the total organic emissions.

15. Page 13, line 2: The individual organics can vary by location and season. Do the organic profiles developed in this work also vary with location, measuring technique, ambient temperature, etc.? In other words, what's the limitation of the developed organic emission profiles when implemented in atmospheric models? Can you add some discussion on limitation in Sect. 5 or in a new section?

16. Page 13, line 10: Based on your statement, the effective SOA yields can vary significantly among different studies. Can you provide a comparative table of the SOA yields derived from the various studies included in this paper in the supplement?

17. Page 14, line 5: Considering the high fraction (as high as 40%-50%) of evaporation for POA even at typical atmospheric conditions, how do you define the phase state for the whole volatility range ($C^*$) in your study?

*Figures and tables:*

18. Figure 1: Can you specify the phase state for each saturation concentration bin at typical atmospheric condition?

19. Figure 3: As illustrated above, comparing the total organic emissions profile (this work) with the gas-phase VOC emission profile (SPECIATE) is not reasonable. Please revise the figure.

---

## Referee Comment (RC2) · Anonymous Referee #2 · 23 Aug 2018

This is a very useful paper that synthesizes motor vehicle emission data and comes up with recommendations for the molecular composition of these emissions to be used in photochemical models and for the volatility distributions to be used in models for secondary organic aerosol (SOA) formation. The results for SOA formation are discussed in terms of existing mechanisms. I recommend that the paper be published after incorporation of the following comments:

The results for the VOC molecular composition obtained in this work are getting short thrift in the presentation. Table S3b gives the lumped composition for use in SAPRC,

but it is not discussed or presented in much detail. I recommend the Authors consider the following changes:

1. Including the detailed VOC composition that went into these lumped compositions would be very helpful in many other analyses and I suggest adding it to Table S3.

2. To draw attention to these results, I suggest the authors include a graph representing the results in Table S3b in the main body of the paper.

3. The Authors would expand the audience for this paper even more, if they included VOC compositions for use in other mechanisms as well, e.g. Carbon-Bond, RACM, GEOS-CHEM.

4. Finally, some discussion of the recommended VOC compositions would be very useful, for example: how do they compare with previous papers, and to what extent does the lumping affect total OH reactivity?

Detailed comments:

Page 5, lines 29-30: Remove "falls"

Page 5, lines 30-31: Remove "falls in"

Page 6, lines 10-11: "estimated" instead of "estimates"

Section 2.3: Equation (2) does not account for the reaction rate coefficients of different compounds. A brief discussion of how this affects the analysis is warranted.

Page 7, line 7: "group" instead of "groups"

Page 11, lines 17-18: But wouldn't this suggest that the enhancement of IVOCs in gasoline exhaust is not the same for different source categories (Pre-LEV vs. ULEV etc.)? That would be in contrast with one of the main messages from this paper.

Figure 3: "cyclic" is consistently misspelled in the legend.

Figure 5, panel a: It is not entirely clear to me what is being plotted here. From the

[Figure]

caption I understand that 1 stands for total NMOG emissions. Is it the mass fraction of NMOGs that is considered an SOA precursor, regardless of the yield? It seems like a very high number.

References: The typesetting made it difficult to distinguish one reference from another.
* * *

---

## Author Comment (AC1) · 23 Nov 2018

We thank both reviewers for their valuable comments. We have updated the manuscript based on their comments and provided a detailed response below. Reviewer comments are in regular black, our response is in blue, and the additions/updated text from the manuscript are in red.

**Reviewer 1**

**Abstract:**

1. Line 18-19: Relatively large variances of the mass fractions for SVOCs are found for the diesel engines (3.1%-17.7%), even using the same fuel type, which means that the end-of-pipe control at least for the diesel engines can effect the mass distribution and cannot be neglected. The sentence of "this suggests that a single profile can be used to represent the emissions from sources operating on the same fuel" is not rigorous and cannot be supported by the main text. Please verify the sentence or specify the conditions.

Author's response: We agree. To revise the manuscript, we have carefully compared the emissions data to define distinct profiles. We have identified five: (i) cold-start and off-road gasoline, (ii) hot-operation gasoline, (iii) gas turbine, (iv) traditional diesel and (v) diesel-particulate-filter equipped diesel.

Changes in the manuscript: In the abstract, adding "This consistency indicates that limited number of profiles are needed to construct emissions inventories. We define five distinct profiles: (i) cold-start and off-road gasoline, (ii) hot-operation gasoline, (iii) gas turbine, (iv) traditional diesel and (v) diesel-particulate-filter equipped diesel."

At the end of section 3.2, adding "An important question is the number of distinct source categories. To investigate this question, Figure 4 compares the volatility distributions of different sources in a 2-dimensional space of IVOC versus SVOC mass fractions. These are important SOA precursors so this framework highlights differences in SOA formation potential. There are three distinct clusters in Figure 4, one for each fuel type (gasoline, diesel and jet). Therefore, these source categories require different profiles. For example, the on-road (cold-start) and off-road gasoline sources emissions cluster, with a median mass IVOC and SVOC fractions of 4.5% and 1.1%, respectively, indicating similar volatility distributions between on- and off-road gasoline sources. Figure 4 also suggests two additional categories, but these distinctions are not as strong given the variability of the data. First, hot-start LDGV emission have much higher IVOC and SVOC fractions than cold-start emissions (18.1% versus 4.5%, 4.7% versus 1.1%). This implies a roughly 4-time higher SOA yield for hot-start on-road gasoline emissions. Therefore, separate profiles should be used to represent cold-start and hot-operation emissions when constructing emission inventories for gasoline vehicles. Second, DPF and non-DPF equipped diesel sources also show significant different volatility distributions, especially in SVOC mass fraction (12.2% versus 3.8%). To account for these differences, we present five emission profiles in Table S3: gasoline (cold-start and off-road), gasoline (hot-start), diesel (non-DPF), diesel (DPF) and gas-turbine engines."

[Figure]

**Figure 4 Two-dimensional visualization of volatility distributions (x axis: IVOC mass fraction, y axis: SVOC mass fraction) of all tested sources. Dashed circles indicate clusters by fuel type: Blue cluster: gasoline (cold-start), navy: gas-turbine, red: non-DPF diesel source.**

**Introduction:**

2. Page 2: The abbreviations of IVOC, SVOC and LVOC should be explained when they appear in the main text for the first time. Please add them.

Author's response: Corrected as suggested.

Changes in the manuscript: Adding "including intermediate-volatile organic compounds (IVOCs, effective saturation concentration $C^*=10^3-10^6$ µg/m$^3$) and semi-volatile organic compounds (SVOCs, $C^*=1-10^2$ µg/m$^3$)"

**Methods:**

3. Since there are several procedures in compilation of the profiles, mapping organics into the VBS, SOA evaluation and also technical details in this section, a diagram illustrating the whole processes would be useful to outline the method more clearly.

Author's response: we added a diagram (Figure S1) illustrating the mapping processes into the VBS.

Changes in the manuscript: Adding "Figure S1 shows the overall processes of mapping speciated and unspeciated compounds data collected on sampling medias to volatility basis set (VBS)."

[Figure]

**Figure S1 Schematic diagram of mapping speciated and unspeciated compounds data to volatility basis set (VBS), $C_n^*$ denotes the C\* value for n-alkanes as surrogate.**

4. Page 3, line 14: You mentioned that "slightly different procedures were used to characterize VOC emissions" confuse me. Can you specify it more clearly? One summary sentence to describe the differences and how much effect on the VOC emissions will help the readers to follow.

Author's response: There are two differences: 1) sampling media and 2) the level of speciation of VOCs compounds. In terms of sampling media, for gasoline and diesel engine exhaust, we used Tedlar bags and canisters for VOC hydrocarbon and carbonyls, respectively, while for gas-turbine exhaust, the total VOC is sampled using SUMMA canisters. We describe the differences in Page 3, line 24 – 27.

"Different levels of speciation were performed on the canister or Tedlar bag samples, depending on source category. The Tedlar bag samples of gasoline exhaust were analyzed for 192 individual VOCs and 10 IVOCs; gas turbine exhaust was analyzed for 81 individual VOCs and 5 IVOCs; diesel exhaust was analyzed for 47 individual VOCs, 2 IVOCs and 11 Kovats lumped groups in the VOC range (organics that has GC retention time between $n_{th}$ and $n+1_{th}$ n-alkanes)."

This effect is addressed to our best effort by "supplementing our gas-turbine and diesel VOC data with existing speciation profiles (SPECIATE profiles #4674 and #5565)" in Page 4 line 28-29. The goal is to combine the existing VOC speciation data with our data, to obtain higher level of speciation.

Changes in the manuscript: We modified this sentence to "slightly different sampling media (Tedlar bags and/or canisters) and level of speciation were used to characterize VOC emissions".

5. Page 4, line 31: As my understanding, the SAPRC mechanism or the CB mechanism are only applied for the gas-phase VOC reactions in atmospheric models. For the organic emissions in the particle phase, they are treated as OM/POA. Please clarify the statement.

Author's response: That is correct. Gas-phase emissions are reported as VOCs in NEI and speciated using either SAPRC or CB mechanism and particle-phase emissions are reported as PM and then speciated as OM/POA. In this paper, we aim to compare the gas-phase emissions using SAPRC mechanism and gas- and particle-phase emission using VBS framework.

Changes in the manuscript: Gas-phase organic emissions must be speciated for use in chemical mechanisms such as SAPRC (Carter, 2010) or Carbon Bond (CB). (…) We compared gas-phase organic emissions using the lumping specified by the SAPRC mechanism; we also compare gas- and particle-phase emissions using the volatility basis set (VBS).

6. Page 5, line 10: Is C* varying with the temperature? The calculated C* in this paper is for T=298K? With the dilution of the vehicle exhaust, the temperature is supposed to decrease gradually to the ambient temperature, to what degree will the C* change during this process? Can you add some illustrations on the variations of C* and phase state of the organic compound with temperature, maybe in the supplement?

Author's response: Yes, C* varies with the temperature. The Clausius-Clapeyron equation is used to calculate the temperature dependence. We use T=298K as the reference temperature, and all C* values in the paper are given at that T. The sensitive of C* to temperature depends on the enthalpy of vaporization. Using the values in CMAQ 5.2 (Murphy et al., 2017), the C* of volatility bin (C* = 1000 μg/m$^3$ at T = 298K) at 273K is 159.7 μg/m$^3$, so a larger fraction will condense at the same OA loading. We added the gas/particle partition curve (X$_p$) in background Figure 1 for quick reference.

Changes in the manuscript: We added the gas/particle partition curve (X$_p$) in background Figure 1 for quick reference. We also added a Table S2 in the supplement for the temperature and OA loading dependence on gas/particle partition curve (X$_p$).

[Figure]

**Figure 1 Volatility distribution of organic emissions for a typical (a) gasoline (b) diesel vehicle. The emissions are classified by sampling media (line 1: Tedlar bag, line 2: bare quartz filter followed by two Tenax tubes)**

**Table S2 Effect of temperature on gas/particle partitioning at equilibrium**

| Log ($C^*$, T = 298K) | $C^*$ (ug/m$^3$) | | | Particle-phase fraction ($X_p$) | | | $\Delta H_{vap}$ (kJ / mol) |
|---|---|---|---|---|---|---|---|
| | T = 298K | T = 273K | T = 320K | T = 298K | T = 273K | T=320K | |
| Nonvolatile | n/a | n/a | n/a | 1.00 | 1.00 | 1.00 | n/a |
| -1 | 1.00E-01 | 3.14E-03 | 1.34E+00 | 0.99 | 1.00 | 0.88 | 96 |
| 0 | 1.00E+00 | 4.72E-02 | 9.85E+00 | 0.91 | 1.00 | 0.50 | 85 |
| 1 | 1.00E+01 | 7.08E-01 | 7.26E+01 | 0.50 | 0.93 | 0.12 | 74 |
| 2 | 1.00E+02 | 1.06E+01 | 5.35E+02 | 0.09 | 0.48 | 0.02 | 63 |
| 3 | 1.00E+03 | 1.60E+02 | 3.94E+03 | 0.01 | 0.06 | 0.00 | 52 |
| 4 | 1.00E+04 | 2.40E+03 | 2.91E+04 | 0.00 | 0.00 | 0.00 | 41 |
| 5 | 1.00E+05 | 3.60E+04 | 2.14E+05 | 0.00 | 0.00 | 0.00 | 30 |
| 6 | 1.00E+06 | 5.41E+05 | 1.58E+06 | 0.00 | 0.00 | 0.00 | 19 |

7. Page 5, line 15: "represents" should be "represent".

Author's response: Corrected as suggested.

8. Page 5, line 27: Please add one or two summary sentences describing the comparison results and conclusions for Figure S1 (a-c).

Author's response: We moved a few sentences from the SI into the main text in this point.

Changes in the manuscript: Adding "For the most volatile of these species, n-pentyl-benzene, the Tedlar bag measured, on average, 5.2 times more than the adsorbent tubes. Both approaches measured essentially the same amount of n-dodecane (ratio of 0.85 and $R^2$ of 0.9. For naphthalene (the least volatile of these species), the adsorbent tubes measured about 5 times more than the Tedlar bag, which we attribute to wall losses in the bag (Wang et al., 1996). "

9. Page 5, line 11: Please explain the abbreviation of "LDGV" when it first appears.

Author's response: Corrected as suggested.

Changes in the manuscript: For all LDGV (light-duty gasoline vehicle) tests, …

**Results and discussion:**

10. Page 7, line 9: Could you provide a table listing the calculated SOA mass yields for IVOCs in the supplement?

Author's response: The calculated IVOC yield values are documented in Table S7 and Table S9 in Zhao et al. (2015). We have added those tables to the supplement (Tables S5, S6).

11. Page 7, line 24: You mentioned that there is also substantial breakthrough of SVOCs from the quartz filter indicated from Figure 1. I may miss some important features in Figure 1. Can you explain more on this "breakthrough" based on Figure 1?

Author's response: Yes. The breakthrough is defined as SVOCs that are not completed absorbed/collected by quartz filter and eventually collected by Tenax tubes. This part of SVOC is breaking through the barrier of quartz filter, and we call it 'breakthrough'. They are denoted as the white bars (Tenax tube) in SVOC range.

Changes in the manuscript: Adding "This breakthrough is denoted by the white bars in the SVOC range in Figure 1; the SVOC breakthrough corresponds to, on average, 37% of the total SVOC emissions from gasoline vehicles, 52% for non-DPF diesel and 89% for DPF-diesel (Zhao et al., 2015, 2016)."

12. Page 9, line 3: The author is suggested to be more careful to use the conjunction words. Since you don't discuss about the phase state for SVOCs or the partition status between gas phase and particle phase for SVOCs in the context, the statement of "the peak in this low-volatility mode is in the middle of the SVOC range" cannot support the following conclusion of "some of the organics are in the particle phase and some of them in the gas phase". More discussions on the partition between gas-particle phases for IVOCs/SVOCs are needed.

Author's response: Thank you for your suggestion! We added a separate paragraph discussing the gas-particle phases for IVOCs/SVOCs.

Changes in the manuscript:

Following the discussion on Figure 1, adding "To illustrate the changes in gas-particle partitioning of IVOCs and SVOCs across a wide range of atmospheric conditions, Figure S2 shows equilibrium particle fraction ($X_p$) for T between 273K – 320K and OA concentration from 1 – 10 μg/m$^3$. IVOCs are essentially exclusively in the gas-phase (> 99%) except at very low temperature (T = 273K) and high OA loading (OA = 10 μg/m$^3$) conditions when about 6% of the lowest bin (C* = 10$^3$ μg/m$^3$) partitions to particle-phase. In contrast, SVOCs are always present in both gas- and particle- phases, in both hot and dilute (T = 320K and OA = 1 μg/m$^3$) or cold and high OA loading (T = 320K and OA = 10 μg/m$^3$) conditions."

[Figure]

**Figure S2 Effect of temperature and OA loading on gas/particle partitioning at equilibrium**

13. Page 9, line 14: Aromatics is also important for this mode, especially for diesel engines.

Author's response: Sorry, we made a mistake in plotting Figure 3, where we wrongly assigned Kovats groups in $C^* = 10^{10}\,\mu g/m^3$ bin to aromatics. The smallest aromatics, benzene ($C^* = 4.24 \times 10^8\,\mu g/m^3$), is in volatility bin of $C^* = 10^9\,\mu g/m^3$, and it only contributes about 5% in that bin. More than 90% of the 'by-product' mode ($C^* = 10^{10}$ -$10^{11}\,\mu g/m^3$ bin) is comprised of smaller alkenes and carbonyls. Thanks for catching this.

Changes in the manuscript:

[Figure]

**Figure 3 Median volatility distribution of organic emissions for (a) cold-start on-road gasoline, (b) gas-turbine (idle) and (c) on-road non-DPF diesel engines. The color shading indicates composition. Shaded area indicated by dashed indicate distribution for unburned fuel; dots indicate traditional source profiles from EPA SPECIATE database. The y-axis has a broken scale to amplify the least volatile emissions.**

14. Page 10, line 13: As my understanding, the traditional profiles are only for the gas-phase VOCs, which were measured using Tedlar bag / canister followed by GC-MS for hydrocarbons, and 2,4-DNPH followed by HPLC for carbonyls. OC is measured and corresponding emissions is assigned in the particle phase. It is not reasonable to compare the mass distributions between "total organic emission" and "gas-phase VOC". Only gas-phase emissions should be included in the comparison, or OC should be also included in the traditional profile to represent the total organic emissions.

Author's response: Yes, this is a very good point. We have included the OC in the traditional profiles in addition to the VOCs. Because OC is modelled as nonvolatile, we assign it in the NV bin. For traditional profiles, the gas- (VOC) and particle-phase (OC) profiles are then renormalized to 1 for proper comparison with our new results.

15. Page 13, line 2: The individual organics can vary by location and season. Do the organic profiles developed in this work also vary with location, measuring technique, ambient temperature, etc.? In other words, what's the limitation of the developed organic emission profiles when implemented in atmospheric models? Can you add some discussion on limitation in Sect. 5 or in a new section?

Author's response: This is correct. All of these measurements were made at summertime in Los Angeles type conditions (~20 °C) using California summertime fuels. We have added a caveat on this point to the manuscript. Changing fuel composition and temperature will alter the composition of the emissions. We will discuss the limitation in Sect. 5.

Changes in the manuscript:

Adding in section 5: "The emission profiles described here (except for gas turbines) are based on experiments conducted using sources recruited from the California in-use fleet, at typical California summertime temperatures (10-25 °C) and using California commercial summertime fuels. Therefore, the data are most directly relevant to California summertime conditions. Ambient temperature can have a large influence on emissions. For example, George et al. (2015) measured about 10 times higher non-methane hydrocarbon (NMHC) emission rates during testing at -7 °C versus 24 °C. The VOC composition also changed with temperature with the fraction of $C_{9+}$ aromatics almost doubling at low temperature. These data suggest that winter emissions may have higher content of larger aromatics ($C_{9+}$ aromatics) and IVOCs, due to incomplete combustion or lower efficiency of catalytic converter. Our profiles therefore likely represent lower bounds to winter vehicle emission in terms of aromatics and IVOC contents. As discussed in Section 3.2, unburned fuel is an important contributor to the emissions. Therefore, variations in fuel composition by, for example, season and/or location will influence the composition of the emissions. From an SOA formation perspective, we are most interested in changes in fuel IVOC and aromatic content. Figure S11 compares our new VOC profiles with data from China (Cao et al., 2016; Yao et al., 2015). There is good agreement for many compounds, but not all.

[Figure]

**Figure S11 Scatter plots of speciated VOC compounds in VOC emission profile to (a) on-road gasoline and (b) on-road diesel vehicle VOC emission in China**

16. Page 13, line 10: Based on your statement, the effective SOA yields can vary significantly among different studies. Can you provide a comparative table of the SOA yields derived from the various studies included in this paper in the supplement?

Author's response: Yes, we included an updated Table 1 with IVOC SOA yield, and will include IVOC mass fraction and total SOA yield in Table S4 in the supplement.

Changes in the manuscript:

**Table 1 List of different estimates of IVOC emissions and SOA yield for mobile sources shown in Figure 6.**

| Label | IVOC Emissions | IVOC SOA yield (at OA = 10 μg/m$^3$) | Reference |
|---|---|---|---|
| **New** | Direct measurements | 0.22 - 0.30 | This work, Zhao et al. (2015, 2016) |
| **Trad** | N/A | N/A | EPA SPECIATE |
| **ROB** | $1.5 \times$ POA | 0.15[a] | Robinson et al. (2007), Koo et al. (2014) |
| **MUR** | $9.656 \times$ POA | N/A | Murphy et al. (2017) |
| **PYE** | $66 \times$ Naphthalene | 0.22 | Pye and Seinfeld (2010) |
| **GEN** | Unburned fuel (1% of NMOG for gasoline, 62% for diesel) | 0.034 - 0.20 | Gentner et al. (2012) |
| **JAT** | Inverting chamber measurements (25% of NMOG for gasoline, 20% of NMOG for diesel) | 0.22 - 0.35 | Jathar et al. (2014) |

[a] From Koo et al. (2014)

**Table S4 Comparison of different estimates of IVOC fraction and overall SOA yield of NMOG emissions**

| | IVOC mass fraction | | | SOA yield | | |
|---|---|---|---|---|---|---|
| | Gasoline | Gas-turbine | Diesel | Gasoline | Gas-turbine | Diesel |
| **This work** | 4.6% | 27.9% | 54.3% | 0.041 | 0.086 | 0.190 |
| **Traditional** | / | / | / | 0.022 | 0.008 | 0.009 |
| **ROB** | 1.2% | 6.1% | 8.0% | 0.023 | 0.018 | 0.021 |
| **MUR** | 7.5% | 39.5% | 51.7% | N/A | N/A | N/A |
| **PYE** | 19.9% | 22.3% | 6.9% | 0.065 | 0.057 | 0.024 |
| **GEN** | 1% | N/A | 62% | 0.023 | N/A | 0.154 |
| **JAT** | 25% | N/A | 20% | 0.071 | N/A | 0.077 |

17. Page 14, line 5: Considering the high fraction (as high as 40%-50%) of evaporation for POA even at typical atmospheric conditions, how do you define the phase state for the whole volatility range (C*) in your study?

Author's response: The profiles intentionally do not directly define phase state as phase state is not a property of the emissions. The issue is that the gas-particle partitioning will vary as atmospheric conditions change (concentrations of organic aerosol and temperature). The profiles specify the volatility distribution of the emissions which can then be used to calculate the gas—particle partitioning (phase state) for any atmospheric conditions. The 40-50% evaporation is from May et al. (2013a and 2013b), which is based on comparing typical CVS dilution sampler conditions to typical atmospheric condition as T = 298K and OA concentration = 10 µg/m$^3$.

Changes in the manuscript:

Adding in the beginning of section 5: "Our new profiles intentionally do not define the phase state of the emissions. Phase state is not a property of the emissions, but determined by the combination of the volatility distribution of the emissions and atmospheric conditions. The issue is that the gas-particle partitioning varies with atmospheric conditions (e.g. concentrations of organic aerosol and temperature). The profiles specify the volatility distribution of the emissions, which can then be used to calculate the gas-particle partitioning (phase state) for any atmospheric condition (Robinson et al., 2010). This approach is critical to correctly predict POA concentrations for sources that have substantial SVOC emissions, such as the sources tested here and biomass smoke (May et al., 2013)"

18. Figure 1: Can you specify the phase state for each saturation concentration bin at typical atmospheric condition?

Author's response: We have added a curve indicating the gas-particle partitioning at T = 298K, OA concentration of 10 µg/ m$^3$ across the entire volatility range to Figure 1, and partitioning coefficients for each saturation concentration bin in Table S4.

19. Figure 3: As illustrated above, comparing the total organic emissions profile (this work) with the gas-phase VOC emission profile (SPECIATE) is not reasonable. Please revise the figure.

Author's response: As described in previous response, we added the nonvolatile OC to the SPECIATE profiles to create a complete traditional profile that accounts for both gas- and particle-phase emissions. We carefully revised the figure to make sure gas-phase emission profile and POA (1.2*OC) sum up to 1 for proper comparison.

**Reviewer 2:**

**General comments on VOC molecular composition:**

Comment 1. Including the detailed VOC composition that went into these lumped compositions would be very helpful in many other analyses and I suggest adding it to Table S3.

Author's response: We now include complete VOC composition in Table S3 for gasoline (cold-start), gasoline (hot-start), gas-turbine, non-DPF and DPF diesel exhaust.

Changes in the manuscript: Our new profiles are the median value of the measured emission for gasoline (separate for cold-start and hot-operations), gas turbine, non-DPF and DPF-diesel sources; they are listed in Table S3 (Supporting Information).

Comment 2. To draw attention to these results, I suggest the authors include a graph representing the results in Table S3b in the main body of the paper.

Author's response: Thank you for your suggestion! The focus of this paper is to present the importance of IVOCs and SVOCs in the new profiles. Our results show that the VOC speciation are well represented in current profiles, but significant efforts are needed to include IVOCs and SVOCs in the profiles.

Comment 3. The Authors would expand the audience for this paper even more, if they included VOC compositions for use in other mechanisms as well, e.g. Carbon-Bond, RACM, GEOS-CHEM.

Author's response: We now included the VOC composition in Table S3 for gasoline (cold-start), gasoline (hot-start), gas-turbine, non-DPF diesel and DPF diesel sources. We also provide the CAS number for each VOC compounds in the profile, which could be used for mapping into different chemical mechanism.

Comment 4. Finally, some discussion of the recommended VOC compositions would be very useful, for example: how do they compare with previous papers, and to what extent does the lumping affect total OH reactivity?

Author's response: We discussed in Page 10, line 19-22: "There is good agreement between our new and traditional profiles in the VOC range, with both having by-product and fuel modes (Fig. 3) and similar chemical composition. Figure 5 demonstrates the strong agreement for SARPC-lumped VOC groups between the new and traditional profiles for all three sources. For example, more than 90% of all SAPRC groups for the gasoline sources agree to within a factor of two."

Changes in the manuscript: Adding to the end of this discussion: "We recommend using our new profiles for VOC composition because they have enhanced VOC speciation from combining the existing SPECIATE profiles with our new experimental data."

**Detailed comments:**

Page 5, lines 29-30: Remove "falls"

Author's response: Corrected as suggested.

Page 5, lines 30-31: Remove "falls in"

Author's response: Corrected as suggested.

Page 6, lines 10-11: "estimated" instead of "estimates"

Author's response: Corrected as suggested.

Section 2.3: Equation (2) does not account for the reaction rate coefficients of different

compounds. A brief discussion of how this affects the analysis is warranted.

Author's response: We are adding a brief discussion here.

Changes in the manuscript: Equation (2) omits the OH reaction rates and therefore represents the ultimate SOA yield from NMOG emissions. The relative contribution of IVOCs and VOCs to SOA varies with time because IVOCs generally react faster with OH than VOCs (Zhao et al., 2016). Therefore, the ultimate yield approach (equation 2) provides a lower bound estimate of the contribution of IVOCs to SOA.

Page 7, line 7: "group" instead of "groups"

Author's response: Corrected as suggested.

Page 11, lines 17-18: But wouldn't this suggest that the enhancement of IVOCs in gasoline exhaust is not the same for different source categories (Pre-LEV vs. ULEV etc.)? That would be in contrast with one of the main messages from this paper.

Author's response: Yes, this is a very good point. There are indeed differences in IVOC enhancement within gasoline sources (Pre-LEV vs. LEV vs. ULEV). However, we hope to emphasize that this difference is largely driven by combustion facility and fuel usage, which gasoline engines are statistically higher in IVOC enhancement than gas-turbine and diesel engines. We added Figure S9, which plots the IVOC enrichment factors of Pre-LEV, LEV and ULEV vehicles exhaust. The figure indicates that the IVOC enhancement is likely influenced by after-treatment/removal. Despite the differences in after-treatment technique, all gasoline engines are statistically higher in IVOC enhancement than gas-turbine and diesel engines. Within gasoline engine source category, we could differentiate the effect of aftertreatment devices. Due to the different removal efficiency between IVOCs and VOCs, median LEV and ULEV vehicles show higher IVOC enrichment factors than pre-LEV vehicles.

Changes in the manuscript:

Adding "There are several possible explanations for this trend. IVOCs may be less efficiently combusted in the engines. Recent research also shows that less IVOCs are removed by catalytic converters compared to VOCs (Pereira et al., 2017). Figure S9 plot the IVOC enrichment factors of Pre-LEV, LEV and ULEV vehicles exhaust. Due to the different removal efficiency between IVOCs and VOCs, median ULEV vehicles show even higher (>10) IVOC enrichment factor."

[Figure]

**Figure S9 IVOC enrichment factors of Pre-LEV, LEV and ULEV vehicles exhaust**

Figure 3: "cyclic" is consistently misspelled in the legend.

Author's response: Corrected as suggested.

Figure 5, panel a: It is not entirely clear to me what is being plotted here. From the caption I understand that 1 stands for total NMOG emissions. Is it the mass fraction of NMOGs that is considered an SOA precursor, regardless of the yield? It seems like a very high number.

Author's response: Yes, panel (a) plot the mass fraction of SOA precursors in total NMOG emissions. We changed the caption to (A) Mass fraction of SOA precursors in total NMOG emissions. Gasoline exhaust show the highest VOC SOA precursor fraction, due to its high aromatic contents in VOC emissions

References: The typesetting made it difficult to distinguish one reference from another.

Author's response: We now change the typesetting to 'Hanging: 0.1' to separate the references.